# Step-Controlled DPO: Leveraging Stepwise Errors for Enhancing Mathematical Reasoning of Language Models

## Abstract

Direct Preference Optimization (DPO) has proven effective at improving the performance of large language models (LLMs) on downstream tasks such as reasoning and alignment. In this work, we propose Step-Controlled DPO (SCDPO), a method for automatically providing stepwise error supervision by creating negative samples of mathematical reasoning rationales that start making errors at a specified step. By applying these samples in DPO training, SCDPO can better align the model to avoid reasoning errors and output accurate reasoning steps. Qualitative analysis of the credit assignment of SCDPO and DPO demonstrates the effectiveness of SCDPO at identifying errors in mathematical solutions. We then apply SCDPO to an InternLM2-20B model, resulting in a 20B model that achieves competitive scores of 88.5% on GSM8K and 58.1% on MATH, rivaling all other open-source LLMs, showing the great potential of our method. The code, models and data are released to inspire future work.

## 1 Introduction

Recently, Direct Preference Optimization (DPO; Rafailov et al. (2024b)) has emerged as a popular choice for aligning large language models (LLMs) with relative feedback to improve the quality of generated text. Prior works Christiano et al. (2023); Pal et al. (2024); Xu et al. (2024) have demonstrated that reinforcement learning algorithms and DPO can improve the mathematical reasoning abilities of LLMs, making the generated reasoning process more controllable. The final answer to a mathematical problem serves as a natural way to judge the quality of the model's response, since a mathematical problem typically has a single correct answer. As a result, the responses producing the correct final answers are desirable and can serve as the preferred samples, while the ones reaching incorrect final answers are undesirable and can serve as the dispreferred samples.

However, solutions to a mathematical problem can be diverse, with many different reasoning paths arriving at the correct final answer and many subtle ways to make mistakes. Determining the preferred and dispreferred responses based on the final answer is coarse and may be inadequate for capturing *the intricacies of the multi-step mathematical reasoning process.* Previous studies introduce process supervision Lightman et al. (2023), but it requires large amounts of meticulous and expensive human annotation and only applies to traditional RL algorithms.

In this paper, we show how to automatically provide explicit stepwise preference supervision by generating diverse dispreferred solutions that start making errors at a specific step. We propose *Step-Controlled DPO (SCDPO)*, an simple yet effective algorithm that introduces stepwise supervision without necessitating extra human annotation. This approach starts with a model finetuned with question-solution pairs and possessing initial math-solving capabilities, which is used to generate solutions to a set of math problems. We choose the solutions whose final answers match those of the ground truth. We take each of these correct solutions and start generating with the model via modulating the hyperparameter of the model, i.e., increasing the temperature of the final softmax function, from various intermediate steps of that solution, and retain the samples where the final answer is incorrect. In this way, the steps before the intermediate step are the same as the original correct solution, while the steps after are the ones with possible errors. During DPO training, the correct solutions are the preferred samples, and they are paired with the wrong solutions generated

054
055
056
057
058
059
060
061
062
063
064
065
066
067
068
069
070
071
072
073
074
075
076
077
078

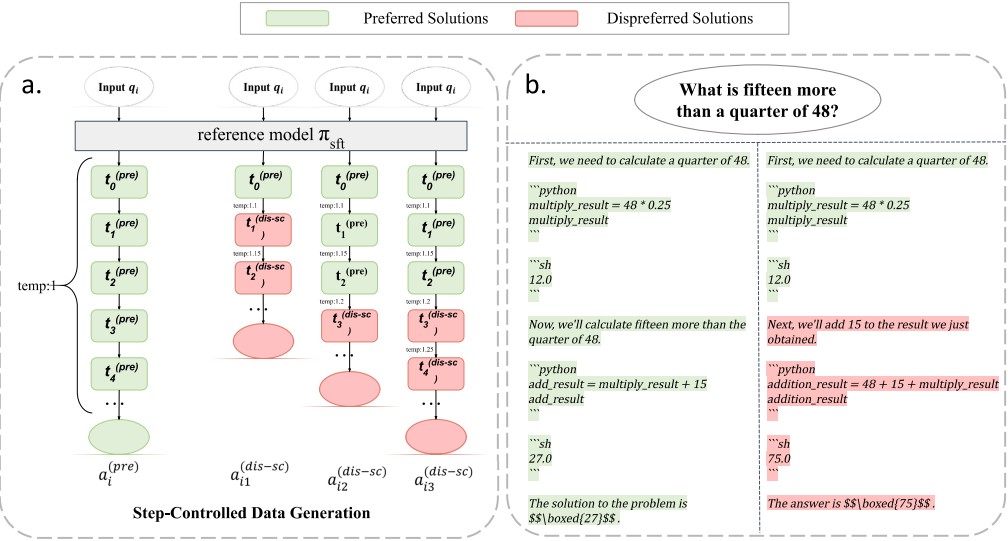

Figure 1: Demonstration and example of the step-controlled data generation process. **a.** Step-controlled data generation. First, a solution reaching the correct final answers is collected, which we denote as $a_i^{(\text{pre})}$. Then, erroneous solutions that reach incorrect final answers are generated, starting from intermediate steps of $a_i^{(\text{pre})}$, creating dispreferred solutions $a_{i1}^{(\text{dis-sc})}$, $a_{i2}^{(\text{dis-sc})}$, and $a_{i3}^{(\text{dis-sc})}$. These dispreferred solutions share the steps before the intermediate steps with $a_i^{(\text{pre})}$. The temperature of the newly generated steps gradually increases with each step to make the generation more erroneous. **b.** An example of a pair of preferred and dispreferred solutions. The dispreferred solution starts making errors after a particular intermediate step.

in this way, with the question and the steps before the intermediate step as the prompts. These step-controlled training samples help models learn detailed reasoning abilities and are mixed with naive DPO training data produced by only checking the final answer, which optimizes the general form of the solution.

Our contributions are as follows:

- We introduce SCDPO, a method that automatically provides explicit stepwise supervision to enhance mathematical abilities of LLMs.

- We conduct pilot experiments on chain-of-thought and code-integrated solutions, showing that SCDPO can effectively improve mathematical problem-solving performance of three different SFT models. We also conduct qualitative analysis of credit assignment of SCDPO.

- Using SCDPO, we finetune an InternLM2-20B model, which reaches 88.5% on GSM8K Cobbe et al. (2021) and 58.1% on MATH Hendrycks et al. (2021), demonstrating the great potential of our method.

## 2 STEP-CONTROLLED DPO PIPELINE

In this section, we introduce Step-Controlled DPO (SCDPO), a pipeline for automatically generating preferred and dispreferred responses to math problems, with annotations of erroneous solving steps, and using these responses in DPO training to enhance the mathematical reasoning abilities of LLMs.

Our method consists of two stages: step-controlled data generation, and step-aware DPO training. The two stages construct a feedback-alignment framework that is both effective and cost-efficient.

**Initial Model.** Our method starts with an initial model, denoted as $\pi_{\text{SFT}}$, which has been finetuned with question-solution pairs from math datasets such as GSM8K and MATH. When prompted with a math problem $q$, $\pi_{\text{SFT}}$ is able to generate a step-by-step solution, denoted as $a$. $a$ can be broken down into a sequence of reasoning steps, for example, $a = (t_0, \ldots, t_m)$. Here, $t_i$ $(i = 0, \ldots, m)$ represents either a code reasoning step or a natural language reasoning step within $a$. For Chain-of-Thought solutions, the reasoning steps are separated by "\n". In code-integrated solutions, the reasoning steps are separated by special tokens as described in Wang et al. (2023a).

## 2.1 STEP-CONTROLLED DATA GENERATION

The data we collect is in two parts: naive DPO data $D_{\text{naive}}$ and Step-Controlled DPO data $D_{\text{SC}}$.

**Generation of $D_{\text{naive}}$.** $D_{\text{naive}}$ contains pairs of preferred-dispreferred samples, used to optimize the general form of the solution. To create $D_{\text{naive}}$, we prompt $\pi_{\text{SFT}}$ with math questions in the training sets of GSM8K and MATH. Each question is presented to $\pi_{\text{SFT}}$ multiple times and various solutions are generated, with a temperature of 1. If a solution reaches the same final answer as the ground truth, and no errors or adjustments occur at any of the reasoning steps (we detect these by looking for strings like "error" or "apologies"), the solution is seen as preferred, while the solutions that reach answers different from the ground truth are considered dispreferred. To find out the frequency of incorrect solution process reaching the correct final answer, we randomly sampled 87 solutions that reach correct final answers, and found that of the 369 reasoning steps in these solutions, only 2 contain errors, which is a very small percentage (about 0.5%). This demonstrates that, in most cases, a correct final answer indicates correct intermediate steps. The solution generation of each question stops when at least one preferred solution and one dispreferred solution are generated, or the number of solutions generated reaches an upper limit of 100. We use questions from the training sets of the GSM8K and MATH datasets for solution sampling, and repeated sampling ensures that 99.8% of the questions in the GSM8K training set and 91.8% of the questions in the MATH training set yield at least one positive sample. The resulting data can be expressed as:

$$D_{\text{naive}} = \{(q_i, a_i^{(\text{pre})}, a_i^{(\text{dis})}) : i = 1, \ldots, N_{\text{naive}}\} \tag{1}$$

Here, $q_i$ denotes the $i$th question, while $a_i^{(\text{pre})}$ and $a_i^{(\text{dis})}$ represent the preferred and dispreferred solution to the $i$th question.

**Generation of $D_{\text{SC}}$.** In order to generate solutions with stepwise error information for DPO training, we propose a method to automatically generate training data with errors starting to occur at a controlled step. The process is demonstrated in Fig. 1. We first take a preferred solution from $D_{\text{naive}}$, denoted as $a_i^{(\text{pre})} = (t_0^{(\text{pre})}, \ldots, t_k^{(\text{pre})}, t_{k+1}^{(\text{pre})}, \ldots, t_{m_i}^{(\text{pre})})$. Here, $t_k^{(\text{pre})}$ is a random intermediate step within $a_i^{(\text{pre})}$. As $a_i^{(\text{pre})}$ is a correct solution, $t_0^{(\text{pre})}, \ldots, t_{m_i}^{(\text{pre})}$ can all be seen as correct steps. As shown in Fig. 1 a, to create a solution with errors occurring after step $k$, we present $\pi_{\text{SFT}}$ with sequence $(q_i, t_0^{(\text{pre})}, \ldots, t_k^{(\text{pre})})$, and raise the temperature of the final softmax function to affect the generation quality, increasing the occurrence of errors in the following steps. Raising the temperature causes the model performance to become unstable and erroneous. We observe that when the temperature is instantly raised and remains at a high value, the model can generate garbled strings as errors accumulate, which does not represent any reasoning mistakes and contains no valuable information. To avoid this, we adopt a gradually increasing temperature, which initially starts at 1.1, and increases by 0.05 with each generated step, until the generation ends or the temperature reaches 1.4. This setting empirically reduces the frequency of the occurrence of garbled text, while increasing the error rate and diversity of generated errors. We generate the steps following $(q_i, t_0^{(\text{pre})}, \ldots, t_k^{(\text{pre})})$ multiple times, until one reaching an incorrect answer is generated. Appending the generated steps to $(t_0^{(\text{pre})}, \ldots, t_k^{(\text{pre})})$, we get a dispreferred solution with step-controlled error, denoted as $a_{ik}^{(\text{dis-sc})} = (t_0^{(\text{pre})}, \ldots, t_k^{(\text{pre})}, t_{k+1}^{(\text{dis-sc})}, \ldots, t_{n_i}^{(\text{dis-sc})})$, where the sequence $(t_{k+1}^{(\text{dis-sc})}, \ldots, t_{n_i}^{(\text{dis-sc})})$ is erroneous. An example is presented in Fig. 1 b. The resulting data can be expressed as:

$$D_{SC} = \{(q_i, a_i^{(pre)}, a_{ik}^{(dis\text{-}sc)}) : i = 1, \ldots, N_{SC}\} \tag{2}$$

Here, $q_i$ denotes the $i$th question, while $a_i^{(pre)}$ is the preferred solution, and $a_{ik}^{(dis\text{-}sc)}$ is the dispreferred solution with step-controlled error that occurs after $t_k^{(pre)}$. $N_{SC}$ is the number of questions in $D_{SC}$, while $m_i$ is the index of the last step of $a_i^{(pre)}$.

## 2.2 STEP-CONTROLLED DPO TRAINING

Having collected $D_{naive}$ and $D_{SC}$, we apply them to DPO training. $D_{naive}$ serves to regulate the general form of solutions, while $D_{SC}$ supervises the model's reasoning on a step level. During DPO training, samples in $D_{naive}$ and $D_{SC}$ are mixed together randomly, and the DPO loss is applied to each sample. For samples from $D_{naive}$, the loss is applied to all steps in the preferred and dispreferred solutions, which can be written as:

$$
\begin{aligned}
&\mathcal{L}_{naive}(\pi_\theta; \pi_{SFT}) \\
&= -\mathbb{E}_{(q_i, a_i^{(pre)}, a_i^{(dis)}) \sim \mathcal{D}_{naive}}[\log \sigma(\beta \log \frac{\pi_\theta(a_i^{(pre)}|q_i)}{\pi_{SFT}(a_i^{(pre)}|q_i)} \\
&\quad - \beta \log \frac{\pi_\theta(a_i^{(dis)}|q_i)}{\pi_{SFT}(a_i^{(dis)}|q_i)})]
\end{aligned}
\tag{3}
$$

For a pair of preferred and dispreferred solutions in $D_{SC}$ where the erroneous steps are generated starting from the $k$th step of the preferred solution, the preferred solution can be denoted as $a_i^{(pre)}$, and the dispreferred solution can be denoted as $a_{ik}^{(dis\text{-}sc)}$. The first $k$ reasoning steps are shared between the pair of solutions. The erroneous steps after the $k$th step in $a_{ik}^{(dis\text{-}sc)}$ is denoted as $(t_{k+1}^{(dis\text{-}sc)}, \ldots, t_{n_i}^{(dis\text{-}sc)})$, while the correct steps in $a_i^{(pre)}$ after the $k$th step is denoted as $(t_{k+1}^{(pre)}, \ldots, t_{m_i}^{(pre)})$. SCDPO directly contrast between the steps in $a_i^{(pre)}$ and $a_{ik}^{(dis\text{-}sc)}$ after the $k$th step, applying the DPO loss only on the different steps.

$$
\begin{aligned}
&\mathcal{L}_{SC}(\pi_\theta; \pi_{SFT}) = \\
&- \mathbb{E}_{(q_i, a_i^{(pre)}, a_{ik}^{(dis\text{-}sc)}) \sim \mathcal{D}_{SC}}[\log \sigma((\sum_{j=k+1}^{m_i} \beta \log \frac{\pi_\theta(t_j^{(pre)}|q_i, t_{<j})}{\pi_{SFT}(t_j^{(pre)}|q_i, t_{<j})}) \\
&- (\sum_{j=k+1}^{n_i} \beta \log \frac{\pi_\theta(t_j^{(dis\text{-}sc)}|q_i, t_{<j})}{\pi_{SFT}(t_j^{(dis\text{-}sc)}|q_i, t_{<j})}))]
\end{aligned}
\tag{4}
$$

Combining $\mathcal{L}_{naive}$ and $\mathcal{L}_{SC}$, the final loss function of Step-Controlled DPO is as follows:

$$\mathcal{L}_{SCDPO} = \mathcal{L}_{naive} + \mathcal{L}_{SC} \tag{5}$$

In this way, $\mathcal{L}_{naive}$ optimizes the general form of the solution, while $\mathcal{L}_{SC}$ focuses on detailed reasoning steps, thus improving the model's accuracy in solving mathematical problems.

## 3 THEORETICAL EXPLANATION OF STEP-CONTROLLED DPO

**Theoretical Insight.** In this section, we provide some theoretical insights into why SCDPO can effectively enhance the reasoning ability of LLMs. As explained in Rafailov et al. (2024a), the DPO loss can be cast into token-level MDP (Markov Decision Process; Puterman (1994)). Similarly,

To find out how many fireflies remained, we need to follow these
steps:
1. Start with the initial count of fireflies: 3.
2. Add 4 less than a dozen more fireflies: \( \text{dozen} = 12 \),
so \( 4 \times \text{(less than a dozen)} = 4 \times (12 - 4) \).
3. Find the total number of fireflies before two flew away.
4. Subtract 2 because two of the fireflies flew away.

To find out how many fireflies remained, we need to follow these
steps:
1. Start with the initial count of fireflies: 3.
2. Add 4 less than a dozen more fireflies: \( \text{dozen} = 12 \),
so \( 4 \times \text{(less than a dozen)} = 4 \times (12 - 4) \).
3. Find the total number of fireflies before two flew away.
4. Subtract 2 because two of the fireflies flew away.

Figure 2: Credit assignment of part of a solution for a GSM8K problem. Each token is colored corresponding to the DPO implicit reward as expressed in Eq. 6 (darker is higher). The left is the credit assignment of SCDPO, which correctly highlights the error – 4 less than a dozen is not 4 times (12 - 4), while the credit assignment of DPO on the right fails to highlight it.

When adding or subtracting numbers, the remainders behave
accordingly. In our case, we're adding the terms, so the overall
remainder will be:

\[
\text{Remainder of sum} = (8 + 1 + 1) \mod 9
\]

When adding or subtracting numbers, the remainders behave
accordingly. In our case, we're adding the terms, so the overall
remainder will be:

\[
\text{Remainder of sum} = (8 + 1 + 1) \mod 9
\]

Figure 3: Credit assignment of part of a solution for a MATH problem. Each token is colored corresponding to the DPO implicit reward as expressed in Eq. 6 (darker is higher). The left is the credit assignment of SCDPO, which correctly highlights the error – as the original question was "Find the remainder when $8 \cdot 10^{18} + 1^{18}$ is divided by 9", the remainders of the terms $8$, $10^{18}$, and $1^{18}$ should not be summed, while the credit assignment of DPO on the right fails to highlight the error.

we can also interpret DPO as a step-level MDP. As presented in Eq. 4, $\beta \log \frac{\pi_\theta(t_j^{(\text{pre})}|q_i, t_{<j})}{\pi_{\text{SFT}}(t_j^{(\text{pre})}|q_i, t_{<j})}$ and $\beta \log \frac{\pi_\theta(t_j^{(\text{dis-sc})}|q_i, t_{<j})}{\pi_{\text{SFT}}(t_j^{(\text{dis-sc})}|q_i, t_{<j})}$ represent the reward of a single preferred or dispreferred step. For naive DPO, all steps in the preferred and dispreferred solutions have their rewards affecting the loss. However, many steps in the dispreferred solution are actually correct, as the error often occurs in a later step. Step-Controlled DPO reduces the range of steps, starting from the $(k+1)$th step, from which the dispreferred steps are more likely to be erroneous due to the raised sampling temperature. The focus of the optimization is thus cast on the errored steps rather than the whole solution, letting the model learn more detailed reasoning abilities.

**Qualitative Evaluation of Credit Assignment of SCDPO.** We perform qualitative evaluation of credit assignment on two models trained with SCDPO and DPO respectively. For a sequence of tokens $\mathbf{x} = (x_0, \ldots, x_m)$, where $x_i$ is the $i$th token in the sequence, we denote all the tokens before $x_i$ as $\mathbf{s}_i$, written as $\mathbf{s}_i = (x_0, \ldots, x_{i-1})$. As introduced in recent research Rafailov et al. (2024a), the DPO implicit reward can be expressed as follows:

$$r(\mathbf{s}_i, x_i) = \beta \log \pi(x_i|\mathbf{s}_i) - \beta \log \pi_{\text{SFT}}(x_i|\mathbf{s}_i) \tag{6}$$

Here $r(\mathbf{s}_i, x_i)$ denotes the DPO implicit reward of token $x_i$, which is the value we visualize as the background color of the token. A darker color represents a higher reward value. As demonstrated in Fig. 2 and Fig. 3, when presented with an incorrect reasoning step, SCDPO more accurately identifies the incorrect tokens compared to DPO. Fig. 2 shows part of a solution for a GSM8K question. In step 2, the solution incorrectly interprets "4 less than a dozen" as "$4 \times (12 - 4)$", when it should have been "$(12 - 4)$". The SCDPO model correctly highlights "$4 \times (12 - 4)$", while the DPO does not. Fig. 3 shows part of a solution for a MATH question. The solution sums the terms in the expression when two of the terms should have been multiplied. SCDPO correctly highlights the incorrect solution, while DPO does not. These examples show that the stepwise supervision provided in SCDPO results in a better token-level understanding of reasoning errors.

| Model | Size | English | | | | | | | Chinese | | |
|-------|------|---------|------|-----|---------|-----------|--------|----------|---------|--------|---------|
| | | GSM8K | MATH | OCW | hung-arian | Mathe-matics | SVA-MP | Simul-eq | APE-210K | CMA-TH | MGSM-zh |
| *Closed-Source Models* | | | | | | | | | | | |
| GPT-3.5 | - | 80.8 | 34.1 | - | 41 | - | - | - | - | 73.8 | - |
| GPT-4 | - | 93.6 | 53.6 | **30.1** | **92** | - | - | - | 84.2 | **89.3** | - |
| GPT-4 Code Interpreter | - | **97.0** | **69.7** | - | - | - | - | - | - | - | - |
| GLM-4 [1] | - | 91.8 | 49.0 | - | 75 | - | - | - | **93.5** | 89.0 | - |
| *Open-Source Models* | | | | | | | | | | | |
| Qwen2 | 7B | 85.7 | 52.9 | 10.7 | 56 | 51.4 | 86.3 | 83.6 | 54.2 | 73.8 | 58.0 |
| Math-Shepherd | 7B | 84.1 | 33.0 | 12.5 | 46 | 36.6 | 81.8 | 84.6 | 45.9 | 68.8 | 67.6 |
| DeepSeekMath-RL | 7B | 86.7 | 58.8 | 22.1 | 55 | 57.4 | 86.7 | 69.6 | 71.9 | 87.6 | 78.4 |
| SVPO | 7B | 81.7 | **59.5** | **34.2** | - | - | - | - | - | - | - |
| InternLM2-Math | 20B | 80.7 | 54.3 | 12.9 | 66 | 41.1 | 83.4 | 55.6 | 64.3 | 69.0 | 58.4 |
| MathGenie | 20B | 87.7 | 55.7 | 23.5 | 69 | 85.1 | 87.3 | 88.5 | - | - | - |
| ChatGLM3-32B | 32B | 82.6 | 40.6 | - | 73 | - | - | - | 89.4 | 85.6 | - |
| ToRA | 34B | 80.7 | 50.8 | 5.5 | - | 77.9 | 80.5 | 50.2 | - | 53.4 | 41.2 |
| MAmmoTH | 70B | 76.9 | 41.8 | - | - | 65.4 | 84.3 | 51.8 | - | - | - |
| MathCoder | 70B | 83.9 | 45.1 | - | - | 74.4 | 84.9 | 77.0 | - | - | - |
| InternLM2-SFT | 20B | 86.4 | 55.8 | 21.6 | 71 | 84.0 | 86.9 | 91.2 | 77.1 | 88.4 | 74.8 |
| InternLM2-SFT-DPO | 20B | 87.0 | 57.6 | 25.5 | 74 | 85.6 | 89.7 | 92.6 | 78.7 | 89.9 | 76.0 |
| InternLM2-SFT-DPO$_{(d\text{-}e)}$ | 20B | 88.2 | 57.5 | 24.5 | 73 | 86.3 | 88.9 | 91.1 | 78.8 | 89.3 | 76.0 |
| InternLM2-SFT-SCDPO | 20B | **88.5** | 58.1 | 29.4 | **78** | **87.5** | **90.2** | 93.6 | 79.3 | **90.3** | **80.4** |

Table 1: Performance of open-source and closed-source models on seven English datasets, GSM8K, MATH, OCW, hungarian, Mathematics, SVAMP and Simuleq, and three Chinese datasets, APE210K, CMATH, and MGSM-zh. All results reported are based on greedy decoding. The best models are marked in **bold**, and the second best models are underlined. Our 20B model trained on SCDPO outperforms SFT and naive DPO on all 10 datasets, demonstrating performance rivaling all other open-source models of similar scales.

# 4 EXPERIMENTS

In this section, we first train a 20B model using SCDPO, reaching a performance rivaling all other models of similar scale. Then, we perform a comprehensive empirical comparison between SCDPO and DPO on three kinds of Mistral-7B SFT models. We also present ablation studies to further explain the design of increasing temperature during the generation of erroneous steps and combining $D_{\text{naive}}$ with $D_{\text{SC}}$ during training.

## 4.1 20B MODEL TRAINED WITH SCDPO

**Training Data.** We collect solutions for questions in the training set of APE210K Zhao et al. (2020), GSM8K and MATH from GPT-4 Code Interpreter. Combining 169K samples from APE210K, 34K from GSM8K and 47K from MATH, we get an SFT dataset of 250K question-solution pairs. The SCDPO and DPO training data is collected as described before in Sec. 2.1. During sampling, top-p is set to 1 and top-k is set to -1 to consider all tokens. The training data for SCDPO contains 13K samples from GSM8K, 46K samples from MATH, and 29K samples from APE210K.

**Training Settings.** We use InternLM2-20B Cai et al. (2024) as the foundation model, as it has demonstrated high performance in previous works Lu et al. (2024); Cai et al. (2024), even surpassing larger models such as Mixtral-8x7B Jiang et al. (2024) and Llama2-70B Touvron et al. (2023) in some cases. In the SFT stage, we finetune the model with a learning rate of $1.0 \times 10^{-5}$ for 3 epochs, with a context length of 2048 tokens. In DPO and SCDPO training, we use a learning rate

| Method | GSM8K | MATH | OCW | hungarian | Mathematics | SVAMP | Simuleq |
|---|---|---|---|---|---|---|---|
| Mistral-7B-Ours | | | | | | | |
| SFT (Baseline) | 76.8 | 43.2 | 21.7 | 52 | 69.8 | 81.3 | 73.9 |
| SFT-continued | 76.3 | 43.9 | 18.8 | 55 | 70.3 | 80.8 | 74.5 |
| SFT+DPO | 78.8 | 45.1 | 18.4 | 56 | 74.8 | 81.0 | 74.9 |
| SFT+DPO$_{(d-e)}$ | 79.0 | 45.7 | 18.0 | 59 | 74.4 | 79.2 | 73.2 |
| SFT+DPO+SC | **80.1** | **47.7** | **22.4** | **61** | **76.5** | **82.3** | **79.0** |
| MetaMath-Mistral-7B | | | | | | | |
| SFT (Baseline) | 77.7 | 28.2 | 12.5 | 33 | 33.9 | 80.0 | 68.5 |
| SFT-continued | 76.8 | 28.5 | 13.2 | 35 | 33.6 | 80.3 | 69.1 |
| SFT+DPO | 81.0 | 28.7 | 14.0 | 34 | 33.8 | 81.0 | 68.3 |
| SFT+DPO$_{(d-e)}$ | 81.4 | 29.0 | 14.7 | 38 | 34.3 | 80.9 | 70.6 |
| SFT+DPO+SC | **81.7** | **29.3** | **15.4** | **42** | **35.0** | **81.6** | **73.2** |
| MathCoder-Mistral-7B | | | | | | | |
| SFT (Baseline) | 78.1 | 39.3 | 12.9 | 62 | 70.4 | 79.4 | 80.5 |
| SFT-continued | 78.2 | 40.3 | 12.5 | 65 | 71.2 | 77.3 | 80.7 |
| SFT+DPO | 79.2 | 42.9 | 14.3 | 65 | 74.9 | 85.4 | 81.3 |
| SFT+DPO$_{(d-e)}$ | 78.3 | 41.1 | 14.7 | 68 | 74.9 | 84.9 | 82.3 |
| SFT+DPO+SC | **80.4** | **43.4** | **15.7** | **70** | **75.4** | **85.4** | **83.1** |

Table 2: Effect of using Step-Controlled DPO (SCDPO) on three different SFT models: Mistral-7B-Ours, MetaMath-Mistral-7B and MathCoder-Mistral-7B. "(d-e)" denote the DPO baseline using the same amount of data as SCDPO. In all three cases, SCDPO outperforms the starting SFT model, continue pretraining on correct samples, naive DPO, and naive DPO with equal amount of data.

| Model | GSM8K | MATH |
|---|---|---|
| Mistral-7B-Ours-SFT | 76.8 | 43.2 |
| Mistral-7B-Ours-SCDPO (temperature=1.0) | 78.6 | 45.9 |
| Mistral-7B-Ours-SCDPO (temperature=1.3) | 80.0 | 45.9 |
| Mistral-7B-Ours-SCDPO (ascending temperature) | **80.1** | **47.7** |

Table 3: Pilot experiments of using different temperatures when generating error steps. When temperature equals 1.0, the errors are not diverse enough. When temperature equals 1.3, the model generates unintelligible strings due to accumulated errors. The design of ascending temperature offers more diversity while avoids generating meaningless errors, resulting in the best performance.

of $1.5 \times 10^{-7}$ to train the SFT model for 2 epochs, with a context length of 1024 and $\beta$ set to 0.1. The models are trained on 16 NVIDIA A800 80GB GPUs with a batch size of 64.

**Evaluation Datasets.** Ten representative mathematical datasets are used in evaluating the models: GSM8K Cobbe et al. (2021), MATH Hendrycks et al. (2021), OCWCourses (OCW) Lewkowycz et al. (2022), Hungarian National Exams (hungarian) Paster (2023), Mathematics Saxton et al. (2019), SVAMP Patel et al. (2021), Simuleq Kushman et al. (2014), APE210K Zhao et al. (2020), CMATH Wei et al. (2023b) and MGSM-zh Shi et al. (2023). The first seven datasets consist of English math questions, while the last three consist of Chinese math questions. The evaluation datasets contain a wide range of problem types, covering mathematical problems from grade-school level to college level, comprehensively evaluating the models' mathematical reasoning abilities. We use greedy decoding for all evaluations.

**Baselines.** We compare our 20B models with powerful closed-source models such as GPT-3.5 (Brown et al., 2020), GPT-4 OpenAI et al. (2024), GPT-4 Code Interpreter OpenAI et al. (2024) and GLM-4 [2], as well as open-source models such as MARIO Liao et al. (2024), Qwen2 Yang et al. (2024), Math-Shepherd Wang et al. (2024), SeaLLM-v2 Nguyen et al. (2024), DeepSeekMath-

---
[2] https://open.bigmodel.cn/dev/api#glm-4

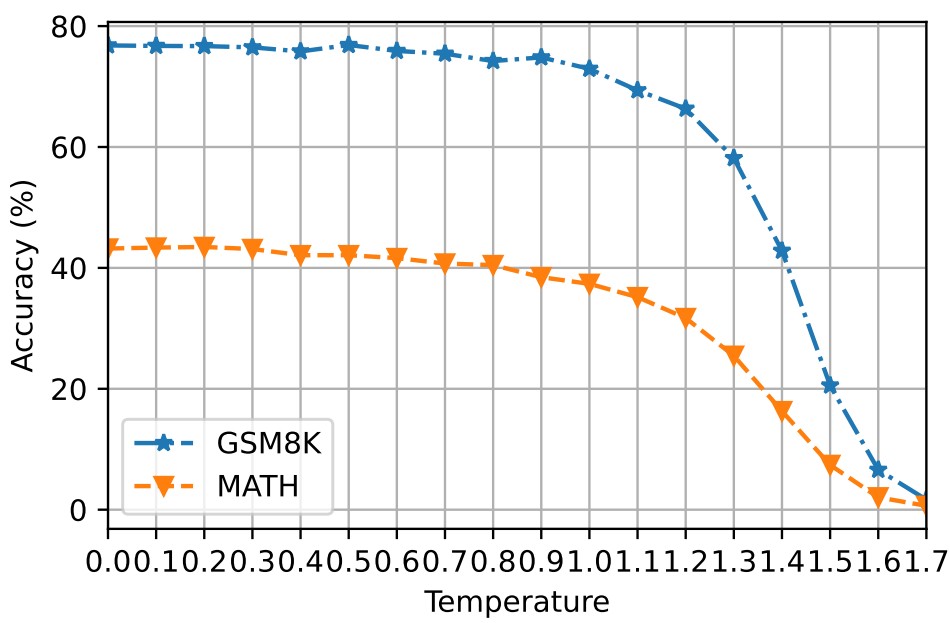

Figure 4: Accuracy of Mistral-7B-Ours (SFT) on GSM8K and MATH when temperature is set at different values.

RL Shao et al. (2024), SVPO Chen et al. (2024), Skywork-13B-Math Yang et al. (2023a), InternLM2-Math [3] Ying et al. (2024), MathGenie Lu et al. (2024), ChatGLM3-32B-RFT-DPO Xu et al. (2024), Yi-Chat Yi (2023), ToRA Gou et al. (2024), MAmmoTH Yue et al. (2023), Math-Coer Wang et al. (2023a) and WizardMath Luo et al. (2023).

**Main Results.** Tab. 1 displays our main results, as well as various closed-source and open-source baselines. Our model achieves a score of 88.5% on GSM8K, 78 on hungarian, 87.5% on Mathematics, 90.2% on SVAMP, 93.6% on Simuleq, 90.3% on CMATH, and 80.4% on MGSM-zh, surpassing all models with published parameters, and obtaining second-best scores among open-source models on APE210K. Our model obtains a score of 58.1% on MATH, which is close to the best and second-best open-source score of 59.5% and 58.8%. While our model rivals the performance of GPT-3.5 on GSM8K and MATH, and surpasses GPT-4 and GLM-4 on MATH, it still underperforms GPT-4 Code Interpreter on GSM8K and MATH, and GLM-4 on APE210K.

Compared to InternLM2-SFT, InternLM2-SFT-SCDPO consistently increases the score on each of the five datasets by approximately 2% to 3%. Compared to both InternLM2-SFT-DPO, which uses the $D_{\text{naive}}$ part of InternLM2-SFT-SCDPO's training data, and InternLM2-SFT-DPO$_{\text{(data-equal)}}$, which uses about the same amount of training data as InternLM2-SFT-SCDPO, InternLM2-SFT-SCDPO consistently achieves the best performance across all five datasets, highlighting the effectiveness of SCDPO in enhancing mathematical problem-solving abilities.

## 4.2 COMPARISON USING DIFFERENT STARTING 7B MODELS

We validate the generalizability of SCDOP on three baseline SFT models: Mistral-7B-Ours, MetaMath-Mistral-7B, and MathCoder-Mistral-7B. Mistral-7B-Ours is finetuned on the 34K GSM8K samples and 47K MATH samples we collected from GPT-4. MetaMath-Mistral-7B is downloaded from the MetaMath HuggingFace repository[4]. MathCoder-Mistral-7B is finetuned using the MathCodeInstruct dataset Wang et al. (2023a), downloaded from HuggingFace[5]. We collect $D_{\text{naive}}$ and $D_{\text{SC}}$ as described in 2.1 using problems from GSM8K and MATH. We compare 4 methods of aligning the starting SFT models: 1. Continue finetuning the starting SFT model using supervised

---

[3] https://github.com/InternLM/InternLM-Math

[4] https://huggingface.co/meta-math/MetaMath-Mistral-7B

[5] https://huggingface.co/datasets/MathLLMs/MathCodeInstruct

| Method | GSM8K | MATH | OCW | hungarian | Mathematics | SVAMP | Simuleq |
|--------|-------|------|-----|-----------|-------------|-------|---------|
| Step-DPO | 80.4 | 29.3 | 12.5 | 42 | 33.4 | 80.4 | 72.6 |
| SCDPO (ours) | **81.7** | 29.3 | **15.4** | 42 | **35.0** | **81.6** | **73.2** |

Table 4: Comparison between our method and Step-DPO on MetaMath-Mistral-7B.

finetuning with preferred solutions from $D_{\text{naive}}$ (SFT-continued). 2. Doing naive DPO training with $D_{\text{naive}}$ (SFT+DPO). 3. Doing naive DPO training with the same amount of training pairs as the SCDPO training, expanded from $D_{\text{naive}}$ (SFT+DPO$_{\text{(d-e)}}$). 4. Doing SCDPO training with $D_{\text{naive}}$ and $D_{\text{SC}}$ (SFT+DPO+SC).

The results are shown in Tab. 2. The purpose of SFT+DPO$_{\text{(d-e)}}$ is to rule out the possibility that the performance gain of SCDPO is the effect of more training samples. SFT-continued shows no obvious gains, likely due to the fact that the models has already been finetuned on many solutions from GSM8K and MATH. As demonstrated in Tab 2, on all three SFT baseline models, SCDPO shows superior performance compared to DPO. This can be attributed to SCDPO's more detailed supervision on the reasoning steps of the math solutions, demonstrating the effectiveness of our method.

We also compare our method with Step-DPO Lai et al. (2024), a work concurrent with ours, which uses GPT-4 to locate erroneous steps. As Step-DPO is in the Chain-of-Thought format, we train MetaMath-Mistral-7B using the dataset and code of Step-DPO. As shown in Tab. 4, our method outperforms Step-DPO on most datasets without relying on any stronger LLMs (e.g., GPT-4), demonstrating the effectiveness of our approach.

### 4.3 Analysis of the Increasing Temperature Design

We present the result of using different temperature during sampling of erroneous steps in Tab. 3. Originally, we tried sampling for the incorrect solutions at the same temperature as the correct solutions (1.0). However, we observed that generated error steps are less diverse than we hoped. Also, as shown in Fig. 4, the accuracy decreases with the increase of temperature, as the generation becomes less stable. We then tried raising the temperature to 1.3, and found that a notable part of the generated solutions contains incomprehensible strings at later steps due to accumulated errors. Finally, we settled on raising the temperature gradually, which enables more diversity while lowering the frequency of generating unintelligible sentences. As shown in Tab. 3, this method also performs best in the pilot experiments.

## 5 Related Work

**LLM for Mathematical Reasoning.** Prior works have explored various methods to enhance mathematical reasoning abilities of LLMs. Prompting methods, such as Chain-of-Thought Wei et al. (2023a), Tree-of-Thought Yao et al. (2023), PAL Gao et al. (2023), Program-of-Thought Chen et al. (2023), and CSV Zhou et al. (2023), use carefully engineered prompts to bring out LLMs' mathematical skills without changing their parameters. Other works optimize parameters of LLMs for enhanced mathematical reasoning through either pretraining or finetuning. Llemma Azerbayev et al. (2024), and MathPile Wang et al. (2023b) continue pretraining LLMs on large amounts of math-related data, while RFT Yuan et al. (2023), Mammoth Yue et al. (2023), MathCoder Wang et al. (2023a), WizardMath Luo et al. (2023), ToRA Gou et al. (2024), MetaMath Yu et al. (2024), MathGLM Yang et al. (2023b), and MathGenie Lu et al. (2024) finetune pretrained models on question-solution pairs. These methods effectively improves LLMs' ability to solve challenging mathematical problems, demonstrating impressive performance on mathematical benchmarks such as GSM8K Cobbe et al. (2021), MATH Hendrycks et al. (2021), etc. Our work builds upon models that have undergone pretraining and finetuning, using DPO to further enhance their mathematical abilities.

**Improving Mathematical Reasoning Using Relative Feedback.** Reinforcement learning from human (or AI) feedback Christiano et al. (2023); Bai et al. (2022) as well as several direct alignment methods Rafailov et al. (2024b); Azar et al. (2023); Zhao et al. (2023); Pal et al. (2024); Ethayarajh

et al. (2024); Liu et al. (2024) have proven effective on various downstream tasks. Our method make use of DPO Rafailov et al. (2024b), introducing a novel way to construct the DPO training data for better enhancement of mathematical abilities of LLMs. Previous works using reinforcement learning or direct alignment methods for improving mathematical reasoning utilize either outcome supervision or process supervision. Outcome supervision such as Shao et al. (2024) is simple and use the outcome of a solution as supervision signal. Lightman et al. (2023) found that process supervision offers better performance than outcome supervision, but needs expert and detailed human or AI annotation, which is difficult to acquire. Math-Shepherd Wang et al. (2023a) and Process Reward Synthesizing Jiao et al. (2024) estimate process rewards with multiple decoding rationales at each step, and train a reward model with the synthesized rewards. Other works such as Xie et al. (2024), Yuan et al. (2024) and Chen et al. (2024) use tree structure to provide fine-grained supervision, often relying on a critique model to decide the correctness of reasoning steps. Concurrent works such as Setlur et al. (2024) and Lai et al. (2024) rely on GPT-4 to synthesize data. Step-DPO (Lai et al., 2024) uses GPT-4 for erroneous step localization, which is less cost-effective. In comparison, our method uses increasing temperature to start generating erroneous steps from intermediate steps of a correct solution, and directly contrast erroneous steps with correct steps, offering a simpler, more cost-effective alternative with high performance, without relying on any stronger LLMs (e.g. GPT-4).

## 6 Limitations and Future Work

Our work contains the following limitations, and we leave them for future work. Firstly, our work is conducted on purely linguistic models, which struggle to solve mathematical problems requiring an understanding of images. Secondly, due to the stepwise attribute of SCDPO, it is not very effective on solution formats consisting of pure code. It only works on solutions consisting of natural language chain of thought or interleaved natural language and code. A method to properly enhance pure code solutions needs to be derived, which we leave for future work. Thirdly, as with all language models, our models can potentially generate hallucinations or produce misleading solutions, which can have a negative effect. Finally, while the data construction method distributionally narrow down the steps likely to be erroneous, it does not indicate the exact step the error occurs, a problem inherent with synthetic process supervision methods. Additionally, our work focuses on mathematical problem-solving, without discussing other reasoning tasks such as code generation, theorem proving, etc. We plan to explore them in future works.

## 7 Conclusion

In this work, we propose Step-Controlled DPO (SCDPO), a method to automatically introduce stepwise error supervision to the process of DPO training by generating dispreferred samples that start making errors at a specified step. SCDPO effectively enhances the mathematical reasoning abilities of LLMs. The 20B model trained with SCDPO on both English and Chinese data achieves high scores on 10 representative mathematical datasets, consistently outperforming naive DPO, demonstrating the effectiveness of our method.

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

| Data | GSM8K | MATH |
|------|-------|------|
| $D_{SC}$ | 79.0% | 46.2% |
| $D_{naive} + D_{SC}$ | 80.1% | 47.7% |

Table 5: Ablation study of using and not using $D_{naive}$ during training. The starting SFT model is Mistral-7B-Ours.

## A    CREDIT ASSIGNMENT ANALYSIS EXAMPLES

In this section, we present several other credit assignment analysis examples, comparing SCDPO to DPO. Fig. 6, Fig. 7 and Fig. 8 show examples of part of the solutions of questions taken from GSM8K and MATH datasets, colored with the DPO implicit reward of each token (darker is higher). As demonstrated in the examples, SCDPO is better than DPO at identifying the errors in the reasoning steps.

## B    ABLATION STUDY OF $D_{NAIVE}$ AND $D_{SC}$

To demonstrate the necessity of combining $D_{naive}$ and $D_{SC}$, we conduct experiment of using only $D_{SC}$ in DPO training. The results are presented in Tab. 5. As demonstrated in the table, combining $D_{naive}$ and $D_{SC}$ results in better performance than only using $D_{SC}$ during DPO training. This is likely because $D_{naive}$ helps regulate the general format of the generated solutions.

## C    ANALYSIS OF GENERATED ERRORS

In this section, we provide quantitative analysis of the erroneous steps generated. We observed seven main kinds of errors: value misusage, condition misinterpretation, coding error, commonsense error, math concept or understanding error, math calculation error, unintelligible strings. The errors are explained as follows:

- Value misusage: misusing values in places where another value should have been used.
- Condition misinterpretation: incorrectly interpreting the meaning or indications of conditions.
- Coding error: making mistakes in code snippets that causes errors.
- Common sense error: misunderstanding of common sense.
- Math concept or understanding error: incorrect recollection or understanding of math concepts.
- Math calculation error: mistakes when making mathematical calculations.
- Unintelligible strings: generation of unintelligible strings that does not represent meaningful reasoning errors.

We randomly sampled 100 incorrect solutions in the training data of SCDPO, and counted the number of each type of error. The result is presented in Fig. 5. As demonstrated in the chart, the reasoning errors generated is diverse, distributed evenly among the different types. Only 4% of the incorrect solutions contain unintelligible strings, demonstrating that the design of gradually increasing temperature can mostly avoid the occurance of meaningless errors.

## D    ERROR RATE OF INTERMEDIATE STEPS WHEN THE FINAL ANSWER IS CORRECT

In this section, we discuss the error rate of intermediate steps in solutions that reaches the correct final answer. As we mentioned in the main paper, we randomly sampled 87 solutions that reach correct final answer, and of the 369 reasoning steps in these solutions, only 2 contain errors, which is a very small percentage (about 0.5%). The 2 erroneous steps are in a question whose answer is to

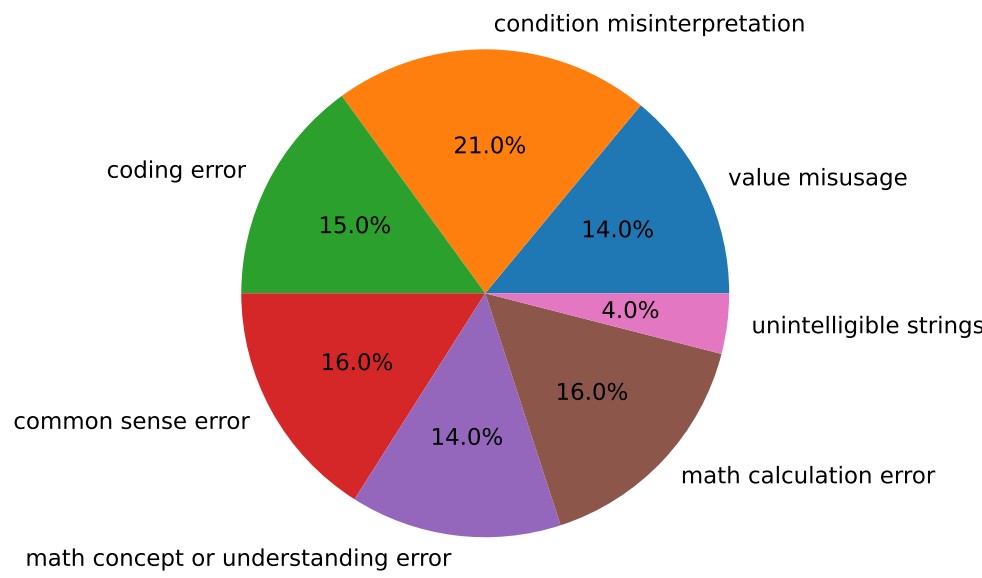

Figure 5: Percentage of each type of error in the 100 examples we sampled.

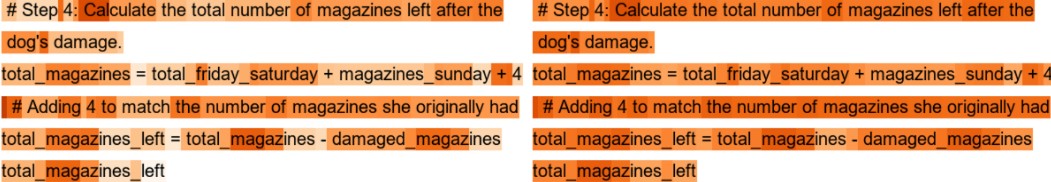

Figure 6: Credit assignment of part of a solution for a GSM8K problem. Each token is colored corresponding to the DPO implicit reward as expressed in Eq. 6 (darker is higher). The left is the credit assignment of SCDPO, which correctly highlighted the error – the number of damaged magazines (which is 4) should not be first added to and then extracted from "total_magazines", while the credit assignment of DPO on the right fails to highlight it.

ask for "the number of real solutions". The possible answers to this question is very limited, which facilitates incorrect reasoning steps to happen to reach a correct final answer. We observe that this kind of question is rare in the GSM8K and MATH datasets we chose to perform SCDPO on, so in most cases a correct final answer indicates that the reasoning steps leading to it is highly likely to be correct. We also removed solutions that contain apologies or error messages, so there are no incorrect steps that are later revised in the correct solutions we retained. As a result, the steps before the generated erroneous steps in SCDPO training data are correct with a high confidence.

## E    DISTRIBUTION OF ERROR GENERATION STARTING POINTS AND ACTUAL ERROR STEP INDEX

We sampled 100 SCDPO rejected solutions and manually inspected them. While not all errors occur at the beginning of the generation of erroneous steps, most of them occur early in the generation, as

```
# Let's denote the number of pencils Antonio has as x.        # Let's denote the number of pencils Antonio has as x.
# According to the problem, Mitchell has x + 6 pencils.        # According to the problem, Mitchell has x + 6 pencils.
# Given that Mitchell has 30 pencils, we can set up the equation:   # Given that Mitchell has 30 pencils, we can set up the equation:

# x + (x + 6) = 30                                            # x + (x + 6) = 30

# Solving for x gives us:                                      # Solving for x gives us:
from sympy import symbols, Eq, solve                           from sympy import symbols, Eq, solve
```

Figure 7: Credit assignment of part of a solution for a GSM8K problem. Each token is colored corresponding to the DPO implicit reward as expressed in Eq. 6 (darker is higher). The left is the credit assignment of SCDPO, which correctly highlighted the error – Mitchell has 30 pencils, and Antonio has 6 less pencils than Michell, which is $30 - 6$, so the introduction of $x$ is not needed, and $x + (x + 6) = 30$ is incorrect, while the credit assignment of DPO on the right fails to highlight it.

```
To solve this problem, we need to find \( n \) such that the sum of   To solve this problem, we need to find \( n \) such that the sum of
an arithmetic series satisfies the given congruence.              an arithmetic series satisfies the given congruence.

The arithmetic series starts at 1 and has a common difference of   The arithmetic series starts at 1 and has a common difference of
5. The last term, 101, can be written as \( 5k \), where \( k = 20 \).   5. The last term, 101, can be written as \( 5k \), where \( k = 20 \).
```

Figure 8: Credit assignment of part of a solution for a MATH problem. Each token is colored corresponding to the DPO implicit reward as expressed in Eq. 6 (darker is higher). The left is the credit assignment of SCDPO, which correctly highlighted the error – $101$ cannot be written as $5k$ where $k = 20$, while the credit assignment of DPO on the right fails to highlight the error.

the earlier steps often make decisions that have a large influence on the correctness of the solution. We present the distribution of the step index where the first actual error occurs in the 100 SCDPO rejected solutions, as well as the distribution of the index of the step where generation of erroneous steps begins. As shown in Tab. 6, the two distributions are closely related, demonstrating that the index of the starting point of error generation have a strong effect on the step index of the actual error. The step index of the actual error is often close to the starting point of the error generation.

We also manipulate the distribution of the step index where error generation begins. Specifically, we train the Mistral-7B-SFT model with SCDPO data where the starting index was either less than or equal to 4, or greater than 4. As shown in Tab. 7, limiting the error starting point index decreases performance compared to not imposing any such limitation.

| Step Index | 2 | 3 | 4 | 5 | 6 | 7 | 8 | 9 | 10 | 11 | 12 | 13 | 14 |
|---|---|---|---|---|---|---|---|---|---|---|---|---|---|
| **Actual Error Distr.** | 17 | 21 | 20 | 12 | 8 | 9 | 5 | 3 | 0 | 4 | 1 | 0 | 0 |
| **Starting Point Distr.** | 32.8 | 23.1 | 15.3 | 11.4 | 6.97 | 4.72 | 2.53 | 1.70 | 0.77 | 0.44 | 0.16 | 0.08 | 0.01 |

Table 6: The distribution of the actual error starting step index in 100 randomly sampled SCDPO rejected solutions and the distribution of the starting point of the error generation of the SCDPO rejected solutions.

| Data | GSM8K | MATH | OCW | Hungarian | Mathematics | SVAMP | Simuleq |
|---|---|---|---|---|---|---|---|
| SFT+DPO+SC | 80.1 | 47.7 | 22.4 | 61 | 76.5 | 82.3 | 79.0 |
| SFT+DPO+SC ($k \leq 4$) | 80.7 | 46.7 | 17.7 | 53 | 74.7 | 82.0 | 79.2 |
| SFT+DPO+SC ($k > 4$) | 79.6 | 46.4 | 17.3 | 59 | 74.1 | 82.3 | 78.6 |

Table 7: Comparison of performance on various datasets using different ranges of error generation starting point during training. "k" is the index of the starting step of error generation. Limiting the starting index slightly affects performance compared to using the full range of indices.

