# OpenReview forum: "Step-Controlled DPO: Leveraging Stepwise Errors for Enhancing Mathematical Reasoning of Language Models"
_ICLR.cc/2025/Conference — Submitted to ICLR 2025_

### Official Review · Reviewer_1hBn · 2024-10-20

**Soundness:** 3
**Presentation:** 3
**Contribution:** 2
**Rating:** 6
**Confidence:** 3

**Summary:**

The paper introduces Step-Controlled DPO, a novel approach to improve the mathematical reasoning abilities of LLMs by leveraging stepwise errors. Unlike Direct Preference Optimization, which relies on final answers to generate preference feedback, SCDPO focuses on stepwise error generation. By creating solutions that introduce errors at specific reasoning steps, SCDPO provides more granular supervision to the model, allowing for better alignment and accuracy in mathematical reasoning. The method is evaluated on several mathematical reasoning benchmarks, showing competitive results with open-source models.

**Strengths:**

1. The SCDPO approach introduces a unique method for improving mathematical reasoning by targeting stepwise errors, which diverges from the traditional focus on outcome-based supervision in DPO.
2. The paper conducts extensive experiments, demonstrating great performance across several mathematical reasoning benchmarks, including GSM8K and MATH.
3. This paper provides clear theoretical explanation of SCDPO, which is necessary and provides theoretical support for this method.
4. This paper is well-structured and clear in its presentation.

**Weaknesses:**

1. The paper mentions SCDPO works well on both CoT and Code-integrated solutions, but the results is not demonstrated this comparison. The author needs to add a table that highlights the comparison between these two different types of solutions.
2. This work lacks of the experiments on show generalisation ability of the method. The author could do an evaluation on out-of-distribution data, comparing the results with the in-distribution data.
3. This work is quite similar to another work "Step-dpo: Step-wise preference optimization for long-chain reasoning of llms", but the differences between these paper are not clearly demonstrated. The authors need to elaborate more about the comparison about this.

**Questions:**

Line 466 misses a citation.

---

> ### Author Response · Authors · 2024-11-21
> **Response to Official Review by Reviewer 1hBn (1/2)**
>
> Thank you for reviewing our work and providing thoughtful suggestions. Your valuable feedback has been instrumental in improving our research and presentation. We address your concerns as follows:
>
> **Q1:** The paper mentions SCDPO works well on both CoT and Code-integrated solutions, but the results do not demonstrate this comparison. The author needs to add a table that highlights the comparison between these two different types of solutions.
>
> **A1:** Thank you so much for the suggestion. We present the comparison in Table 2 of the paper, which has been expanded to include results on out-of-domain datasets. MetaMath-Mistral-7B uses CoT solutions, while Mistral-7B-Ours and MathCoder-Mistral-7B utilize Code-Integrated solutions. To better compare these two types of solutions, we evaluated these models on five out-of-domain datasets: OCW, Hungarian, Mathematics, SVAMP, and Simuleq.
> The results are shown below. Code-Integrated solutions outperform CoT solutions on benchmarks involving complex computations, such as MATH, OCW, and Mathematics. On less computation-intensive datasets, such as GSM8K, SVAMP, and Simuleq, the performance of the two types of solutions is similar.
>
> |**Method**|**GSM8K**|**MATH**|**OCW**|**Hungarian**|**Mathematics**|**SVAMP**|**Simuleq**|
> |-|-|-|-|-|-|-|-|
> |**Mistral-7B-Ours**|||||||||
> |SFT(Baseline)|76.8|43.2|21.7|52|69.8|81.3|73.9|
> |SFT-continued|76.3|43.9|18.8|55|70.3|80.8|74.5|
> |SFT+DPO|78.8|45.1|18.4|56|74.8|81.0|74.9|
> |SFT+DPO(d-e)|79.0|45.7|18.0|59|74.4|79.2|73.2|
> |**SFT+DPO+SC**|**80.1**|**47.7**|**22.4**|**61**|**76.5**|**82.3**|**79.0**|
> |**MetaMath-Mistral-7B**|||||||||
> |SFT(Baseline)|77.7|28.2|12.5|33|33.9|80.0|68.5|
> |SFT-continued|76.8|28.5|13.2|35|33.6|80.3|69.1|
> |SFT+DPO|81.0|28.7|14.0|34|33.8|81.0|68.3|
> |SFT+DPO(d-e)|81.4|29.0|14.7|38|34.3|80.9|70.6|
> |**SFT+DPO+SC**|**81.7**|**29.3**|**15.4**|**42**|**35.0**|**81.6**|**73.2**|
> |**MathCoder-Mistral-7B**|||||||||
> |SFT(Baseline)|78.1|39.3|12.9|62|70.4|79.4|80.5|
> |SFT-continued|78.2|40.3|12.5|65|71.2|77.3|80.7|
> |SFT+DPO|79.2|42.9|14.3|65|74.9|85.4|81.3|
> |SFT+DPO(d-e)|78.3|41.1|14.7|68|74.9|84.9|82.3|
> |**SFT+DPO+SC**|**80.4**|**43.4**|**15.7**|**70**|**75.4**|**85.4**|**83.1**|
>
>
> **Q2:** This work lacks the experiments to show generalization ability of the method. The author could do an evaluation on out-of-distribution data, comparing the results with the in-distribution data.
>
> **A2:** Thank you for the suggestion. Firstly, as shown in Table 1 and the table below, we tested our models on 10 datasets, among which GSM8K, MATH, and APE210K are used for training and are considered in-distribution. **The other 7 are out-of-distribution datasets**. OCW Courses is a dataset of challenging university-level problems. The Hungarian National Exam consists of held-out high school national final math exams, making it an excellent benchmark for testing the model’s generalizability. Mathematics is a dataset focused on solving complex equations, while SVAMP and Simuleq feature math problems that are worded differently from those in GSM8K. These datasets cover a broad range of domains and problem types.
>
> Secondly, we also tested SCDPO’s effect on three different SFT models with out-distribution datasets, as shown in the table in A1. We have added these new tests to Table 2 in the revised paper. These results can effectively demonstrate the generalization ability of the method.
>
> | Model                   | Size | GSM8K  | MATH   | OCW   | Hungarian | Mathematics | SVAMP  | Simuleq | APE210K | CMATH  | MGSM-zh |
> |-------------------------|------|--------|--------|-------|-----------|-------------|--------|---------|---------|--------|---------|
> | InternLM2-SFT          | 20B  | 86.4   | 55.8   | 21.6  | 71        | 84.0        | 86.9   | 91.2    | 77.1    | 88.4   | 74.8    |
> | InternLM2-SFT-DPO      | 20B  | 87.0   | 57.6   | 25.5  | 74        | 85.6        | 89.7   | 92.6    | 78.7    | 89.9   | 76.0    |
> | InternLM2-SFT-DPO(d-e) | 20B  | 88.2   | 57.5   | 24.5  | 73        | 86.3        | 88.9   | 91.1    | 78.8    | 89.3   | 76.0    |
> | InternLM2-SFT-SCDPO    | 20B  | **88.5** | **58.1** | **29.4** | **78**    | **87.5**    | **90.2** | **93.6**  | **79.3**  | **90.3** | **80.4** |

---

> ### Author Response · Authors · 2024-11-21
> **Response to Official Review by Reviewer 1hBn (2/2)**
>
> **Q3:** This work is quite similar to another work "Step-dpo: Step-wise preference optimization for long-chain reasoning of llms", but the differences between these papers are not clearly demonstrated. The authors need to elaborate more about the comparison.
>
> **A3:** Thank you for your valuable suggestion. Step-DPO [1] is a work concurrent with ours, which was also submitted to ICLR 2025 (https://openreview.net/forum?id=H5FUVj0vMd). The approach in [1] uses GPT-4 for erroneous step localization, which is less cost-effective. In contrast, our method incorporates stepwise information into the DPO process by constructing positive and negative pairs that differ after a specific step.
>
> To compare Step-DPO with our method, we trained MetaMath-Mistral-7B using the dataset and code of Step-DPO. As shown in the table below, our method outperforms Step-DPO on most datasets without relying on any stronger LLMs (e.g., GPT-4), demonstrating the effectiveness of our approach. We have addressed the differences between our work and the concurrent Step-DPO in the related work section (line 485, blue font) and added the experimental results in Section 4.2.
>
> | MetaMath-Mistral-7B          | GSM8K  | MATH  | OCW   | Hungarian | Mathematics | SVAMP  | Simuleq |
> |----------------|--------|-------|-------|-----------|-------------|--------|---------|
> | SFT (Baseline) | 77.7   | 28.2  | 12.5  | 33.0      | 33.9        | 80.0   | 68.5    |
> | Step-DPO       | 80.4   | 29.3  | 12.5  | 42        | 33.4        | 80.4   | 72.6    |
> | SCDPO (ours)   | **81.7** | 29.3 | **15.4** | 42      | **35.0** | **81.6** | **73.2** |
>
>
> [1] Step-DPO: Step-wise Preference Optimization for Long-chain Reasoning of LLMs
>
>
> **Q4:** Line 466 misses a citation.
>
> **A4:** Thank you for pointing this out. We have corrected this typo in the revised paper. Additionally, we have thoroughly proofread the entire paper again to ensure that any other typos or errors have been addressed. We sincerely appreciate your valuable feedback.

---

> ### Author Response · Authors · 2024-11-23
> **Looking forward to your reply**
>
> Dear Reviewer 1hBn,
>
> We hope this message finds you well. As the author-reviewer discussion period progresses, we respectfully seek your confirmation on the adequacy of our rebuttal in addressing the concerns raised in your review.
>
> We really appreciate the substantial time and effort you have committed to reviewing our work and are grateful for the additional insights. Your comments have been very helpful in refining our project.
>
> Thank you once again for your valuable perspectives. We eagerly await your further guidance.
>
> Sincerely,
> Authors

---

> > ### Comment · Reviewer_1hBn · 2024-11-24
> >
> > Thanks for your detailed explanation. I have raised my score.

---

> > > ### Author Response · Authors · 2024-11-24
> > > **Thanks for your response.**
> > >
> > > Dear Reviewer 1hBn,
> > >
> > > Thank you so much for taking the time to review our work and for acknowledging our response and efforts. We truly appreciate your thoughtful feedback and guidance.
> > >
> > > Sincerely,
> > > The Authors

---

### Official Review · Reviewer_qGoa · 2024-11-04

**Soundness:** 3
**Presentation:** 3
**Contribution:** 3
**Rating:** 6
**Confidence:** 4

**Summary:**

This work proposes SCDPO, a method leveraging stepwise supervision without human annotation to enhance mathematical reasoning in models. The approach uses a model that is trained in math domains to generate both positive and negative samples. To produce dispreferred samples, the model is prompted with the initial portion of correct reasoning chains, then asked to complete the reasoning at a high temperature setting, leading to potential errors in the final answer. In the SCDPO loss, only the latter part of the reasoning chains is considered. Empirical results demonstrate that models trained with SCDPO outperform those trained with other methods.

**Strengths:**

1. The paper is well-written and easy to follow.
2. The method for generating erroneous steps in SCDPO is both interesting and novel. It provides an effective approach to achieve process-based supervision without human annotation on intermediate steps. This approach allows for error localization within certain steps, which—while not precise—represents a step towards finer-grained, automatic process-based supervision.
3. The experiments are solid and convincing, the proposed method demonstrates strength across different models and benchmarks.

**Weaknesses:**

1. The concept of a “step” in the reasoning chain is not clearly defined in the paper. It would be more clear if the authors could provide a more precise explanation of how they segment steps within a reasoning chain.
2. Potential positional bias in error generation. The approach uses an ascending temperature strategy to create erroneous data, which may unintentionally introduce a bias where errors occur more frequently in the latter half of the reasoning chain. It would be helpful if the authors could investigate whether the distribution of errors indeed skews toward the end.
Since errors cannot be automatically located within the chain, manually sampling and inspecting some dispreferred solutions could provide insights into any potential biases.

**Questions:**

1. (Following up on Weakness 2) Could the authors provide an analysis of the distribution of the dispreferred data? Specifically:

a. What is the distribution of the step $k$ (referenced in Line 138) that is used as the starting point for generating errors in the resulting dispreferred data?

b. What is the distribution of the "actual error" location within the dispreferred reasoning chains?

2. Line 467: There appears to be an unlinked reference.

---

> ### Author Response · Authors · 2024-11-21
> **Response to Official Review by Reviewer qGoa**
>
> Thank you for reviewing our work and for your thoughtful suggestions, which have greatly helped improve our project. We address your concerns below:
>
> **Q1:** The concept of a “step” in the reasoning chain is not clearly defined in the paper. It would be more clear if the authors could provide a more precise explanation of how they segment steps within a reasoning chain.
>
> **A1:** Thank you for your suggestion. In CoT-format solutions, the reasoning steps are separated by “\n”. In Code-Integrated format solutions, the model generates reasoning steps in the following order: a natural language reasoning step, a code snippet, the output of the code execution, and then repeats this cycle. Each block begins with `<|text|>`, `<|code|>` or `<|execution|>` and ends with `<|end_of_text|>`. We consider each text block as a reasoning step, and each code snippet paired with its execution output is also regarded as a reasoning step.
>
> **Q2:** Potential positional bias in error generation. The approach uses an ascending temperature strategy to create erroneous data, which may unintentionally introduce a bias where errors occur more frequently in the latter half of the reasoning chain. It would be helpful if the authors could investigate whether the distribution of errors indeed skews toward the end. Since errors cannot be automatically located within the chain, manually sampling and inspecting some dispreferred solutions could provide insights into any potential biases.
>
> **A2:** Thank you for your suggestion. We sampled 100 SCDPO-rejected solutions and manually inspected them. Our findings indicate that while not all errors occur at the beginning of the generation of erroneous steps, most occur early in the process, as earlier steps often make decisions that significantly influence the correctness of the solution.
> We present the distribution of the step index where the first actual error occurs in the 100 SCDPO-rejected solutions, alongside the distribution of the index where the generation of erroneous steps begins. As shown in the table below, the two distributions are closely related, demonstrating that the starting point of error generation has a strong impact on the step index of the actual error. In most cases, the step index of the actual error is close to the starting point of error generation. This analysis has been added to Appendix E of the revised paper.
>
> |step index|2|3|4|5|6|7|8|9|10|11|12|13|14|
> |-|-|-|-|-|-|-|-|-|-|-|-|-|-|
> |actual error distribution|17|21|20|12|8|9|5|3|0|4|1|0|0|
> |starting point distribution|32.84|23.13|15.25|11.39|6.97|4.72|2.53|1.7|0.77|0.44|0.16|0.08|0.01|
>
>
> **Q3:** (Following up on Weakness 2) Could the authors provide an analysis of the distribution of the dispreferred data? Specifically:
>
> (a) What is the distribution of the step (referenced in Line 138) that is used as the starting point for generating errors in the resulting dispreferred data?
>
> (b) What is the distribution of the "actual error" location within the dispreferred reasoning chains?
>
> **A3:** We present (a) and (b) in the table below. (a) corresponds to the “starting point distribution”, while (b) corresponds to the “actual error distribution”. As shown in the table, the two distributions are closely related, with the starting point of error generation having a strong influence on the location of the actual error.
>
> |step index|2|3|4|5|6|7|8|9|10|11|12|13|14|
> |-|-|-|-|-|-|-|-|-|-|-|-|-|-|
> |actual error distribution|17|21|20|12|8|9|5|3|0|4|1|0|0|
> |starting point distribution|32.84|23.13|15.25|11.39|6.97|4.72|2.53|1.7|0.77|0.44|0.16|0.08|0.01|
>
>
> **Q4:** Line 467: There appears to be an unlinked reference.
>
> **A4:** Thank you for pointing out this typo. We have corrected it in the revised paper. Additionally, we have thoroughly proofread the entire paper again to address any other potential errors. We sincerely appreciate your attention to detail and your valuable feedback.

---

> > ### Comment · Reviewer_qGoa · 2024-11-25
> >
> > Thank you for your detailed response and for providing additional analysis. Incorporating the explanation would further enhance the clarity of the paper. I will maintain my original score.

---

> > > ### Author Response · Authors · 2024-11-25
> > > **Thank you for your response.**
> > >
> > > Dear Reviewer qGoa,
> > >
> > > We sincerely appreciate you taking the time to review our work and acknowledge our response and efforts. Your thoughtful feedback and guidance mean a great deal to us.
> > >
> > > Thank you again for the time and effort you have dedicated to reviewing our work.
> > >
> > > Best regards,
> > > The Authors

---

### Official Review · Reviewer_buSo · 2024-11-04

**Soundness:** 3
**Presentation:** 3
**Contribution:** 3
**Rating:** 8
**Confidence:** 4

**Summary:**

The authors propose Step-Controlled DPO (SCDPO), which, different from DPO, provides fine-grained, step-wise training signals by creating negative samples of mathematical reasoning rationales with errors after a specified step. The authors conduct experiments using InternLM2 and Mistral customized versions, showing better performance across seen and unseen datasets, compared to SFT and DPO baselines. They also did ablations to understand the impact of temperature on negative sample generation and the benefits of integrating naive DPO with SCDPO.

**Strengths:**

1. The proposed methodology provides fine-grained, step-wise training signals, which is missing in most DPO and RFT work done in mathematical reasoning task.
2. The explanation of the 2-step pipeline is straightforward and clear.
3. The authors conduct experiments using various models including InternLM2 and Mistral, showing better and generalizable performance  compared to SFT/DPO baselines and various open-source and closed-source models with zero-shot prompting. They also did ablations to understand the impact of temperature on negative sample generation and the benefits of integrating naive DPO with SCDPO.

**Weaknesses:**

1. In section 4.2, the authors validate the generalizability of SCDOP on three baseline SFT models, but only show performance on the datasets that have been used for SCDOP training. It would be good to also show performance on the datasets that are not used for training, similar to table 1, to better understand its generalizability not only across models but also across datasets.
2. The paper mentions that they determine the correctness of reasoning steps by matching the final answer with the ground truth. However, it is not clear how they do the matching. Since the matching mechanism cannot be perfect as the same answer could be in different expressions, e.g. "increasing from 4 to 5" is equivalent to "increasing by 25%", it would be good to add quantitative or qualitative analysis of the matching strategy. In the other words, the paper currently only contains analysis on false negatives but not on false positives.
3. [Minor point] Typos:
    a. line 121: "We" -> "we"
    b. figure 1.a, the order of i3 and i2 should be switched
    c. line 366: 78 -> 78%

**Questions:**

1. Since it's not guaranteed to have errors at the first generated step, as mentioned in the limitation, it would be good to add quantitative and qualitative analysis on which step the model generates error, to understand the granularity of the training signals. Additionally, it would be interesting to see if manipulating the distribution of the step index it generates error would cause any difference in performance.
2. Mathematical reasoning is one of reasoning types. It would interesting to see if it can expand to other reasoning tasks as well.
3. There are many "-" in table 1. Any reason?

---

> ### Author Response · Authors · 2024-11-21
> **Response to Official Review by Reviewer buSo (1/4)**
>
> Thank you for taking the time and effort to review our work. We sincerely appreciate your thoughtful suggestions, which have been invaluable in helping us improve our project. We have addressed your concerns below:
>
> **Q1:** In section 4.2, the authors validate the generalizability of SCDPO on three baseline SFT models, but only show performance on the datasets that have been used for SCDPO training. It would be good to also show performance on the datasets that are not used for training, similar to table 1, to better understand its generalizability not only across models but also across datasets.
>
> **A1:** Thank you for your thoughtful suggestion. We have added the results of out-of-domain datasets to the experiments on three baseline SFT models. As shown in the table below, SCDPO consistently improves performance across all five out-of-domain datasets. These results have been incorporated into **Table 2 of the revised paper**. We greatly appreciate your feedback, which has helped enhance the clarity and comprehensiveness of our work.
>
> | **Method**            | **GSM8K** | **MATH** | **OCW** | **Hungarian** | **Mathematics** | **SVAMP** | **Simuleq** |
> |------------------------|-----------|----------|---------|---------------|-----------------|-----------|-------------|
> | **Mistral-7B-Ours**    |           |          |         |               |                 |           |             |
> | SFT (Baseline)         | 76.8      | 43.2     | 21.7    | 52            | 69.8            | 81.3      | 73.9        |
> | SFT-continued          | 76.3      | 43.9     | 18.8    | 55            | 70.3            | 80.8      | 74.5        |
> | SFT+DPO                | 78.8      | 45.1     | 18.4    | 56            | 74.8            | 81.0      | 74.9        |
> | SFT+DPO (d-e)          | 79.0      | 45.7     | 18.0    | 59            | 74.4            | 79.2      | 73.2        |
> | **SFT+DPO+SC**         | **80.1**  | **47.7** | **22.4**| **61**        | **76.5**        | **82.3**  | **79.0**    |
> | **MetaMath-Mistral-7B**|           |          |         |               |                 |           |             |
> | SFT (Baseline)         | 77.7      | 28.2     | 12.5    | 33            | 33.9            | 80.0      | 68.5        |
> | SFT-continued          | 76.8      | 28.5     | 13.2    | 35            | 33.6            | 80.3      | 69.1        |
> | SFT+DPO                | 81.0      | 28.7     | 14.0    | 34            | 33.8            | 81.0      | 68.3        |
> | SFT+DPO (d-e)          | 81.4      | 29.0     | 14.7    | 38            | 34.3            | 80.9      | 70.6        |
> | **SFT+DPO+SC**         | **81.7**  | **29.3** | **15.4**| **42**        | **35.0**        | **81.6**  | **73.2**    |
> | **MathCoder-Mistral-7B**|          |          |         |               |                 |           |             |
> | SFT (Baseline)         | 78.1      | 39.3     | 12.9    | 62            | 70.4            | 79.4      | 80.5        |
> | SFT-continued          | 78.2      | 40.3     | 12.5    | 65            | 71.2            | 77.3      | 80.7        |
> | SFT+DPO                | 79.2      | 42.9     | 14.3    | 65            | 74.9            | 85.4      | 81.3        |
> | SFT+DPO (d-e)          | 78.3      | 41.1     | 14.7    | 68            | 74.9            | 84.9      | 82.3        |
> | **SFT+DPO+SC**         | **80.4**  | **43.4** | **15.7**| **70**        | **75.4**        | **85.4**  | **83.1**    |
>
>
> **Q2:** The paper mentions that they determine the correctness of reasoning steps by matching the final answer with the ground truth. However, it is not clear how they do the matching. Since the matching mechanism cannot be perfect as the same answer could be in different expressions, e.g. "increasing from 4 to 5" is equivalent to "increasing by 25%", it would be good to add quantitative or qualitative analysis of the matching strategy. In the other words, the paper currently only contains analysis on false negatives but not on false positives.
>
> **A2:** Thank you for your valuable suggestion. During the SFT process, the ground truth answer is appended to the end of each training sample as `<|text|>ground truth answer<|end_of_block|>`. This helps standardize the format of the generated solutions and improves the accuracy of data extraction and matching. Additionally, we use SymPy to evaluate expressions, ensuring that the matching strategy is highly accurate. To verify this, we randomly sampled 100 solutions and manually inspected the matching accuracy. We did not identify any false positives or false negatives. Since most of the answers in GSM8K and MATH are numerical, their correctness can be easily verified against the ground truth.

---

> ### Author Response · Authors · 2024-11-21
> **Response to Official Review by Reviewer buSo (2/4)**
>
> **Q3:** [Minor point] Typos: a. line 121: "We" -> "we" b. figure 1.a, the order of i3 and i2 should be switched c. line 366: 78 -> 78%
>
> **A3:** Thank you for pointing out these typos. We have corrected the typos in line 121 and Figure 1.a in the revised paper. The "78" in line 366 should not be followed by a "%" as it represents the score the model achieved in the Hungarian exam. This score is calculated by summing the points for each question the model answered correctly. We have also thoroughly proofread the paper multiple times to address any remaining typos. We sincerely appreciate the time, effort, and valuable insights you have provided in reviewing our paper.
>
>
> **Q4:** Since it's not guaranteed to have errors at the first generated step, as mentioned in the limitation, it would be good to add quantitative and qualitative analysis on which step the model generates error, to understand the granularity of the training signals. Additionally, it would be interesting to see if manipulating the distribution of the step index it generates error would cause any difference in performance.
>
> **A4:** Thank you for your suggestion. We sampled 100 SCDPO rejected solutions and manually inspected them. Our findings indicate that while not all errors occur at the very beginning of the generation of erroneous steps, most occur early in the process. This is because earlier steps often make decisions that significantly influence the correctness of the solution.
> We present the distribution of the step index where the first actual error occurs in the 100 SCDPO rejected solutions, alongside the distribution of the index where the generation of erroneous steps begins. As shown in the table below, the two distributions are closely related, demonstrating that the starting point of error generation has a strong effect on the step index of the actual error. In most cases, the step index of the actual error is close to the starting point of error generation. This analysis has been added to Appendix E of the revised paper.
>
> |step index|2|3|4|5|6|7|8|9|10|11|12|13|14|
> |-|-|-|-|-|-|-|-|-|-|-|-|-|-|
> |actual error distribution|17|21|20|12|8|9|5|3|0|4|1|0|0|
> |starting point distribution|32.84|23.13|15.25|11.39|6.97|4.72|2.53|1.7|0.77|0.44|0.16|0.08|0.01|
>
> The following are two examples. In **Example 1**, the actual error occurs at the starting point of error generation. The model incorrectly generates the line of code `dolphins_in_training = 2 * fully_trained_dolphins // 3`, where the `fully_trained_dolphins` should have been `still_to_be_trained_dolphins`. In **Example 2**, the actual error occurs at the fourth step after the starting point of error generation. The model incorrectly outputs, “Starting logs + logs burned - logs added = total logs after 3 hours.”

---

> ### Author Response · Authors · 2024-11-21
> **Response to Official Review by Reviewer buSo (3/4)**
>
> **Example 1:**
>
> Question:
> There are 20 dolphins in the aqua park. One-fourth of the dolphins are fully trained. Two-third of the remaining dolphins are currently in training and the rest will be trained next month. How many dolphins will be trained next month?
>
> Correct Steps:
>
> To solve this problem, let's proceed step by step:
>
> Step 1: Find the number of fully trained dolphins
> To find out how many dolphins are fully trained, we can calculate one-fourth of the 20 dolphins in the aqua park.
>
> The formula for calculating one-fourth of a number $ n $ is:
>
> $$
> \text{Fully trained dolphins} = \frac{n}{4}
> $$
>
> In this case, $ n = 20 $.
>
> Let's calculate.
>
> ```python
> # Given number of total dolphins in the aqua park
> total_dolphins = 20
>
> # Calculate the number of fully trained dolphins
> fully_trained_dolphins = total_dolphins // 4
> fully_trained_dolphins
> ```
> ```output
> 5
> ```
>
> The number of fully trained dolphins in the aqua park is 5.
>
> Step 2: Calculate the number of dolphins still to be trained
> The number of still-to-be-trained dolphins can be found by subtracting the number of fully trained dolphins from the total number of dolphins:
>
> $$
> \text{Dolphins still to be trained} = \text{Total dolphins} - \text{Fully trained dolphins}
> $$
>
> Let's calculate this.
>
> ```python
> # Calculate the number of dolphins that are still to be trained
> still_to_be_trained_dolphins = total_dolphins - fully_trained_dolphins
> still_to_be_trained_dolphins
> ```
> ```output
> 15
> ```
>
> The number of dolphins that are still to be trained is 15.
>
> Step 3: Find the number of dolphins currently in training
> To find out how many dolphins are currently in training, we take two-thirds of the number of still-to-be-trained dolphins.
>
> The formula for this is:
>
> $$
> \text{Dolphins in training} = \frac{2}{3} \times \text{Dolphins still to be trained}
> $$
>
> Let's calculate this next.
>
> Erroneous Steps:
>
> ```python
> # Calculate the number of dolphins that are currently in training
> dolphins_in_training = 2 * fully_trained_dolphins // 3
> dolphins_in_training
> ```
> ```output
> 3
> ```
>
> The number of dolphins that are currently in training is 3.
>
> Step 4: Calculate the number of dolphins that will be trained next month
> The number of dolphins that will be trained next month can be calculated by subtracting the number of dolphins currently in training from the number of remaining dolphins that are still to be trained:
>
> $$
> \text{Dolphins for next month} = \text{Still to be trained dolphins} - \text{Dolphins in training}
> $$
>
> Let's perform this calculation.
>
> ```python
> # Calculate the number of dolphins that will be trained next month
> dolphins_next_month = still_to_be_trained_dolphins - dolphins_in_training
> dolphins_next_month
> ```
> ```output
> 12
> ```
>
> $$
> \boxed{12}
> $$
>
> The number of dolphins that will be trained next month is $\boxed{12}$.
>
> **Example 2:**
>
> Question:
> A cookfire burns three logs every hour. It was built with six logs to start. If it gets two more logs added to it at the end of every hour, how many logs will be left after 3 hours?
>
> Correct Steps:
>
> Alright, let's break this problem down step by step.
>
> 1. Calculate the number of logs burned in 3 hours.
> 2. Calculate the number of logs added to the fire in 3 hours.
> 3. Subtract the number of logs burned from the initial number of logs and the number of logs added to find out how many logs will be left after 3 hours.
>
> Let's begin with the first step:
>
> Step 1: Calculate the number of logs burned in 3 hours.
>
> The cookfire burns 3 logs every hour. So for 3 hours:
> Number of logs burned = $ 3 \times \text{number of hours} $
>
> Erroneous Steps:
>
> ```python
> # Given data
> logs_burned_per_hour = 3
> hours = 3
>
> # Step 1: Calculate the number of logs burned in 3 hours
> logs_burned_in_3_hours = logs_burned_per_hour * hours
> logs_burned_in_3_hours
> ```
> ```output
> 9
> ```
>
> In 3 hours, $9$ logs will be burned.
>
> Step 2: Calculate the number of logs added to the fire in 3 hours.
>
> If 2 logs are added to the fire at the end of every hour, then in 3 hours:
> Number of logs added = $2 \times \text{number of hours}$
>
> ```python
> # Given data for the number of logs added each hour
> logs_added_per_hour = 2
>
> # Step 2: Calculate the number of logs added to the fire in 3 hours
> logs_added_in_3_hours = logs_added_per_hour * hours
> logs_added_in_3_hours
> ```
> ```output
> 6
> ```
>
> In 3 hours, $6$ logs will be added to the fire.
>
> Step 3: Calculate the total number of logs after 3 hours.
>
> Starting logs + logs burned - logs added = total logs after 3 hours,Where:
> Starting logs = 6
>
> Therefore, total logs after 3 hours = $6 + 9 - 6$
>
> Let's compute that.
>
> ```python
> # Initial starting logs
> initial_starting_logs = 6
>
> # Step 3: Calculate the total number of logs after 3 hours
> total_logs_after_3_hours = initial_starting_logs + logs_burned_in_3_hours - logs_added_in_3_hours
> total_logs_after_3_hours
> ```
> ```output
> 9
> ```
>
> After 3 hours, there will be $\boxed{9}$ logs left.

---

> ### Author Response · Authors · 2024-11-21
> **Response to Official Review by Reviewer buSo (4/4)**
>
> **A4 (continued):**
> We also tested manipulating the distribution of the step index where error generation begins. Specifically, we trained the Mistral-7B-SFT model with SCDPO data where the starting index was either less than or equal to 4, or greater than 4. As shown in the table below, limiting the error starting point index decreases performance compared to not imposing any such limitation. We have added this result to Appendix E of the revised paper.
>
> | Data                | GSM8K | MATH | OCW  | Hungarian | Mathematics | SVAMP | Simuleq |
> |---------------------|-------|------|------|-----------|-------------|-------|---------|
> | SFT+DPO+SC          | 80.1  | 47.7 | 22.4 | 61        | 76.5        | 82.3  | 79.0    |
> | SFT+DPO+SC (k <= 4) | 80.7  | 46.7 | 17.7 | 53        | 74.7        | 82.0  | 79.2    |
> | SFT+DPO+SC (k > 4)  | 79.6  | 46.4 | 17.3 | 59        | 74.1        | 82.3  | 78.6    |
>
>
> **Q5:** Mathematical reasoning is one of reasoning types. It would interesting to see if it can expand to other reasoning tasks as well.
>
> **A5:** Thank you for the valuable suggestion. In this paper, we mostly focus on mathematical problem-solving tasks as it is a representative reasoning task and can be easily verified. We will explore the effect of SCDPO on other reasoning tasks such as code generation, theorem proving, and other STEM subjects in future works. We have added this to the Limitation and Future Work Section of our paper.
>
> We test our 20B models on GPQA Diamond, a challenging multiple-choice Q&A dataset featuring difficult questions in biology, physics, and chemistry. As shown in the table below, SCDPO also improves performance on GPQA, demonstrating the potential of our method for reasoning tasks in subjects beyond mathematics.
>
> | InternLM2-20B             | GPQA  |
> |-------------------|-------|
> | SFT              | 26.9  |
> | SFT+DPO          | 27.3  |
> | SFT+DPO(e-q)     | 26.8  |
> | SFT+DPO+SC       | 28.3  |
>
> **Q6:** There are many "-" in table 1. Any reason?
>
> **A6:** The performance of the baselines is taken from the reports in previous works ([1], [2], [3], [4], [5]). However, some datasets were not included in these works. To make the comparison in Table 1 clearer, we conducted our own evaluation of several important open-source baseline models on these previously unreported datasets. The results are as follows, and we have added them to Table 1 of the revised paper.
>
> |Models	|GSM8K	|MATH	|OCW	|hungarian	|Mathematics	|SVAMP	|Simuleq	|Ape210K	|CMATH	|MGSM_zh|
> |---|---|---|---|---|---|---|---|---|---|---|
> |Qwen2-7B	|85.7	|52.9	|10.7	|56	|51.4	|86.3	|83.6	|54.2	|73.8	|58|
> |Math-Shepherd-7B	|84.1	|33	|12.5	|46	|36.6	|81.8	|84.6	|45.9	|68.8	|67.6|
> |DeepSeekMath-RL-7B	|86.7	|58.8	|22.1	|55	|57.4	|86.7	|69.6	|71.9	|87.6	|78.4|
> |MathGenie-20B	|87.7	|55.7	|23.5	|69	|85.1	|87.3	|88.5	|	|||
> |InternLM2-Math-20B	|80.7	|54.3	|12.9	|66	|41.1	|83.4	|55.6	|64.3	|69.0	|58.4|
>
> [1] Xu, Yifan, et al. "ChatGLM-Math: Improving Math Problem-Solving in Large Language Models with a Self-Critique Pipeline." arXiv preprint arXiv:2404.02893 (2024).
> [2] Gou, Zhibin, et al. "Tora: A tool-integrated reasoning agent for mathematical problem solving." arXiv preprint arXiv:2309.17452 (2023).
> [3] Wang, Ke, et al. "Mathcoder: Seamless code integration in llms for enhanced mathematical reasoning." arXiv preprint arXiv:2310.03731 (2023).
> [4] Yue, Xiang, et al. "Mammoth: Building math generalist models through hybrid instruction tuning." arXiv preprint arXiv:2309.05653 (2023).
> [5] Lu, Zimu, et al. "Mathgenie: Generating synthetic data with question back-translation for enhancing mathematical reasoning of llms." arXiv preprint arXiv:2402.16352 (2024).
> Experiment: testing

---

> > ### Comment · Reviewer_buSo · 2024-11-28
> >
> > Thanks for the clarification and more experimental details. I will keep my positive score

---

> > > ### Author Response · Authors · 2024-11-30
> > > **Thank you for your response.**
> > >
> > > Dear Reviewer buSo,
> > >
> > > Thank you for taking the time to review our work and acknowledge our response and efforts. We sincerely appreciate your thoughtful feedback and guidance, which are invaluable to us.
> > >
> > > Once again, thank you for your time and dedication in evaluating our work.
> > >
> > > Best regards,
> > > The Authors

---

### Official Review · Reviewer_b65V · 2024-11-06

**Soundness:** 3
**Presentation:** 2
**Contribution:** 2
**Rating:** 5
**Confidence:** 3

**Summary:**

The paper presents Step-Controlled Direct Preference Optimization (SCDPO), a method designed to enhance the mathematical reasoning capabilities of large language models (LLMs). SCDPO extends Direct Preference Optimization (DPO) by incorporating step-by-step error supervision data. The negative (dispreferred) samples in SCDPO is generated by model itself by adjusting the model's sampling temperature until reasoning mistakes are introduced at specific steps.  SCDPO was tested on the InternLM2-20B and Mistral-7B, achieving competitive performance on benchmark datasets like GSM8K and MATH.

**Strengths:**

- The motivation of this paper that including step supervision in DPO for solving math problem is straightforward but reasonable.

- The step supervision is efficiently introduced by self-generation of LLMs, reducing the cost of annotation.

- The authors have demonstrated the effectiveness of the proposed method comparing to baselines on some regular benchmarks such as GSM8K and MATH

**Weaknesses:**

- Ths most concern is the lack of many baselines (that are kind of resembling the proposed method) in experiments. There are some previous works, e.g., Step-DPO [1], introduce the idea of step supervision in DPO for enhancing the math capabilities of LLMs. This work does make some differences by using temperature to controll the step errors, the comparison should be made with previous step-supervised SFT/DPO methods to demonstrate the effectiveness.

[1] Step-DPO: Step-wise Preference Optimization for Long-chain Reasoning of LLMs


- The idea of using softmax temperature to controll LLMs to make errors at certain steps is interesting but kind of limited. The naive problems can be accurately solved by LLM step by step, while the authors carefully craft the decoding temperature to "mislead" the LLMs to generate wrong answer at certain steps. Although the step supervision can be collected between the right and wrong answers, the LLM seems to hardly be improved as it can naively solve the problem and the step supervision just reduces some potential errors it may make from random decoding. In other words, the LLM is naively easier to give the right answer but harder to give the wrong answer (need to tune the temperature), making it doubtful if the LLM can achieve much improvment from such step supervision. Have you tried using a larger LLM to generate the step supervision data to improve smaller LLMs? It may be better than the self-improvement of smaller LLMs.

- The paper writing and structure can be improved. Some important informations are missed in this paper,  for example, it's unclear what the hyperparameters (top-p, top-k) used for generating training data, and how many samples the final training data contains. The cite of  Process Reward Synthesizing in related work is also missed.

**Questions:**

See weaknesses above.

---

> ### Author Response · Authors · 2024-11-21
> **Response to Official Review by Reviewer b65V (1/2)**
>
> Thank you for taking the time and effort to review our work. Your suggestions have been invaluable in helping us improve our project. We address your concerns below:
>
> **Q1:** The most concern is the lack of many baselines (that are kind of resembling the proposed method) in experiments. There are some previous works, e.g., Step-DPO [1], that introduce the idea of step supervision in DPO for enhancing the math capabilities of LLMs. This work does make some differences by using temperature to control the step errors, the comparison should be made with previous step-supervised SFT/DPO methods to demonstrate the effectiveness.
>
> **A1:** Thank you so much for your comment.
>
> Step-DPO [1] is a work **concurrent with ours**, which was also submitted to **ICLR 2025**(https://openreview.net/forum?id=H5FUVj0vMd). The approach in [1] uses GPT-4 for erroneous step localization, which is less cost-effective. In contrast, our method incorporates stepwise information into the DPO process by constructing positive and negative pairs that differ after a specific step. Our method can build the positive and negative pairs by the model itself without incurring expensive costs from closed-source LLMs (e.g., GPT-4), which is especially advantageous for large-scale data scenarios.
>
> To compare Step-DPO with our method, we trained MetaMath-Mistral-7B using the dataset and code of Step-DPO. As shown in the table below, our method outperforms Step-DPO on most datasets, demonstrating the effectiveness of our approach. We have addressed the differences between our work and the concurrent Step-DPO in the related work section (line 485, blue font) and added the experimental results in Section 4.2.
>
>
> | MetaMath-Mistral-7B          | GSM8K  | MATH  | OCW   | Hungarian | Mathematics | SVAMP  | Simuleq |
> |----------------|--------|-------|-------|-----------|-------------|--------|---------|
> | SFT (Baseline) | 77.7   | 28.2  | 12.5  | 33.0      | 33.9        | 80.0   | 68.5    |
> | Step-DPO       | 80.4   | 29.3  | 12.5  | 42        | 33.4        | 80.4   | 72.6    |
> | SCDPO (ours)   | **81.7** | 29.3 | **15.4** | 42      | **35.0** | **81.6** | **73.2** |
>
>
>
> [1] Lai, Xin, et al. "Step-dpo: Step-wise preference optimization for long-chain reasoning of llms." arXiv preprint arXiv:2406.18629 (2024).
>
>
> **Q2:** The idea of using softmax temperature to control LLMs to make errors at certain steps is interesting but kind of limited. The naive problems can be accurately solved by LLM step by step, while the authors carefully craft the decoding temperature to "mislead" the LLMs to generate wrong answers at certain steps. Although the step supervision can be collected between the right and wrong answers, the LLM seems to hardly be improved as it can naively solve the problem and the step supervision just reduces some potential errors it may make from random decoding. In other words, the LLM is naively easier to give the right answer but harder to give the wrong answer (need to tune the temperature), making it doubtful if the LLM can achieve much improvement from such step supervision. Have you tried using a larger LLM to generate the step supervision data to improve smaller LLMs? It may be better than the self-improvement of smaller LLMs.
>
> **A2:** Thank you so much for your question. While the right answers are sampled from the LLM itself, they are retained from 100 times of sampling. In many cases, the LLM reaches the correct answer only after many failed attempts. For example, during the first round of sampling, only 61.01% of the questions in the MATH dataset were solved correctly.
>
> SCDPO provides supervision to enhance the LLM’s reasoning ability and increases the likelihood of the model reaching the correct answer on the first try, thereby improving its overall accuracy. Utilizing samples generated by the model itself for SCDPO demonstrates that our method enables the model to self-improve, showcasing both the effectiveness and efficiency of our approach.
>
> To demonstrate the effect of training a smaller model using SCDPO data generated by a larger model, we utilized the SCDPO data produced by our InternLM2-SFT-20B model to train Mistral-7B-Ours. As shown in the table below, training with SCDPO data generated by the 20B model does not consistently improve performance across all datasets. This may be due to differences in the output distributions of the 7B and 20B models.
>
> | Data                  | GSM8K | MATH | OCW  | Hungarian | Mathematics | SVAMP | Simuleq |
> |-----------------------|-------|------|------|-----------|-------------|-------|---------|
> | Generated-with-7B-self | 80.1  | **47.7** |**22.4** | **61**      | **76.5**        | 82.3  | 79.0    |
> | Generated-with-20B-model | **80.7**  | 46.8 | 18.4 | 58        | 73.8        | **85.1**  | **81.1**    |

---

> ### Author Response · Authors · 2024-11-21
> **Response to Official Review by Reviewer b65V (2/2)**
>
> **Q3:** The paper writing and structure can be improved. Some important information is missing in this paper, for example, it's unclear what the hyperparameters (top-p, top-k) used for generating training data, and how many samples the final training data contains. The cite of Process Reward Synthesizing in related work is also missed.
>
> **A3:** Thank you for your valuable suggestion. During sampling, we set top-p to 1 and top-k to -1 to consider all tokens. The final training data for the 20B model includes 13K samples from GSM8K, 46K samples from MATH, and 29K samples from APE210K. We have added these important details, as well as the previously missing citation for Process Reward Synthesizing, to the revised paper. Additionally, we have carefully proofread the paper multiple times to address any remaining typographical errors. We sincerely appreciate the time, effort, and invaluable insights you have provided in reviewing our paper.

---

> > ### Comment · Reviewer_b65V · 2024-11-21
> > **Response to Authors**
> >
> > - Step-DPO was released 5 months ago and I do believe it should be included in the baselines out of the pretty similar intuition. Now it is good to see the positive results of your proposed method.
> >
> > - I got the motivation. However,  my concern is that SCDPO can only help LLM to approach potential right answer paths, which can be natively accessed by itself, with larger probabilities. So the method may be limited to harder questions that are non-trivial to be solved by small LLMs. Hence, I recommend considering integrating larger LLMs to generate the step supervision data to cut the edge of smaller LLMs. The new results may show that the 20B model cannot help a lot for 7B models, but I guess larger models in the same family (with identical vocabulary) may be more useful.
> >
> > - Good to see the revisions.
> >
> > Thank the authors for the new results that address my first concern (comparison with Step-DPO). But I still a bit concerned about the potential of SCDPO for challenging tasks (e.g., the improvements on MATH are less significant than that on GSM-8K). Generally, I can raise my rating to 5.

---

> > > ### Author Response · Authors · 2024-11-21
> > > **Thank you very much for your response.**
> > >
> > > Thank you for acknowledging our motivation and the effort we put into our rebuttal. We greatly appreciate the time and effort you dedicated to reviewing our work. Your insights have been invaluable in improving our project.
> > >
> > > Regarding the suggestion to integrate larger LLMs to generate step supervision data to enhance the performance of smaller LLMs, it is possible that larger models in the same family may be more effective and could further improve the performance of smaller models. To validate this, we are using Mixtral-8x7B to generate SCDPO data to supervise Mistral-7B. However, due to limited computational resources, the process is taking a significant amount of time. We will include the results in future updates. On the other hand, the fact that data generated by the model itself can effectively improve its own performance already demonstrates the effectiveness of SCDPO. Thank you again for your insightful suggestion.
> > >
> > > To address your concern about the potential of SCDPO for challenging tasks, we present the results of our 20B models on GPQA Diamond, a challenging graduate-level, Google-proof Q&A dataset featuring difficult questions in biology, physics, and chemistry. We tested these results for Q5 of Reviewer buSo. As shown in the table below, SCDPO also improves performance on GPQA, demonstrating the potential of our method for challenging reasoning tasks in subjects beyond mathematics. We plan to expand SCDPO to even more challenging tasks in the future.
> > >
> > > | InternLM2-20B             | GPQA  |
> > > |-------------------|-------|
> > > | SFT              | 26.9  |
> > > | SFT+DPO          | 27.3  |
> > > | SFT+DPO(e-q)     | 26.8  |
> > > | SFT+DPO+SC       | 28.3  |
> > >
> > > Thank you once again for acknowledging our motivation and rebuttal efforts. We truly appreciate your thoughtful suggestions and valuable insights.

---

> ### Author Response · Authors · 2024-11-29
> **Integrating larger LLMs to generate the step supervision data**
>
> To address the suggestion to integrate larger LLMs for generating step supervision data to enhance the capabilities of smaller LLMs, we generated SCDPO data using Mistral-8x7B to train Mistral-7B-Ours. As shown in the table below, using a larger LLM from the same family improves performance across most datasets. We will add the results to the revised paper.
>
> On the other hand, as demonstrated in the paper, using self-generated data consistently enhances performance across both in-domain and out-of-domain datasets. This is because, even with a smaller model, 99.8% of the questions in the GSM8K training set and 91.8% of the questions in the MATH training set yielded at least one positive sample after 100 sampling attempts. This improved performance from self-generated data highlights SCDPO’s ability to help the model self-improve, further demonstrating the effectiveness of our method.
>
> | Data                     | GSM8K | MATH  | OCW  | Hungarian | Mathematics | SVAMP  | Simuleq |
> |--------------------------|-------|-------|------|-----------|-------------|--------|---------|
> | Generated-with-7B-self   | 80.1  | 47.7  | 22.4 | 61        | 76.5        | 82.3   | 79.0    |
> | Generated-with-Mistral-8x7B | **80.7** | **47.8** | **23.1** | **61**      | **77.5**      | **85.3** | **83.3** |
>
> Thank you once again for your insightful suggestion. If you have any further questions or suggestions, please feel free to let us know!

---

> > ### Comment · Reviewer_b65V · 2024-12-03
> > **Response to Authors**
> >
> > Dear Authors,
> >
> > Thank you for your comprehensive rebuttal. While I appreciate the additional results and clarifications provided, I still have one remaining concern regarding the core motivation of your approach - improving LLMs through training on self-solved questions with step-by-step supervision.
> >
> > The new results demonstrating Mixtral-8x7B's ability to guide Mistral-7B are particularly promising, suggesting that your method may have significant potential when applied in a knowledge distillation framework. This finding opens up interesting possibilities for model enhancement. However, this point is not the primary target of the current version, which is underexplored in the main text with enough discussions. I believe the paper would have great potential to be improved by incorporating more content regarding this point.  Generally, I have raised my rating from 3 to 5, but have no enough confidence to give a higher rating.

---

> > > ### Author Response · Authors · 2024-12-03
> > > **Thank you for your response.**
> > >
> > > Dear Reviewer b65V,
> > >
> > > Thank you for your insightful comments and acknowledgment of our paper’s motivation and our rebuttal efforts.
> > >
> > > The extensive experiments in our paper **already demonstrate that SCDPO consistently improves performance** compared to SFT and naive DPO on various in-domain and out-of-domain datasets, **highlighting the effectiveness of SCDPO**. This fully supports our motivation to introduce step-wise supervision in DPO training by constructing positive and negative samples that differ at an intermediate step.
> > >
> > > Integrating larger LLMs to generate step supervision data is an interesting and constructive addition to our work, and **we will incorporate the suggested experiments into the revised paper**. Thank you once again for your insightful suggestion.
> > >
> > > Best wishes,
> > > The Authors

---

### Official Review · Reviewer_5qbR · 2024-11-09

**Soundness:** 3
**Presentation:** 3
**Contribution:** 2
**Rating:** 5
**Confidence:** 3

**Summary:**

This paper proposes Step-Controlled DPO, a new data-curating methodology that improves the effectiveness of DPO fine-tuning on reasoning tasks. When DPO is applied to fine-tune LLMs on reasoning datasets, one naive way of curating the dataset is to sample pairs of correct/wrong reasoning chains from LLMs. The authors of this work make the observation that when LLMs generate incorrect reasoning chains, they usually start making mistakes after a specific position; in particular, a reasoning chain is incorrect does not mean that the whole chain is incorrect, and its prefix could potentially be correct. Hence, the authors propose to generate negative examples by taking a correct reasoning chain and resample its suffix starting from some position k, which they refer to as SC-DPO. The authors combine the naively generated data with the data generated from SC-DPO, and conduct DPO fine-tuning. Compared to the baseline where only naively generated data is used, their approach show consistent improvement across different models and datasets.

**Strengths:**

The approach presented is simple, effective and immediately applicable to a wide range of logical/mathematical reasoning benchmarks.

**Weaknesses:**

The contribution overall seems relatively minor: SC-DPO is not really a new fine-tuning objective but instead the original DPO objective combined with a rather specific way of generating negative examples; the empirical gains are also relatively small.

Most of the information from Table 1. is probably not super important, e.g. why do the authors need to compare the performance of InternLM2-SFT-SCDPO against the listed open-source LLMs? how is this comparison related to SC-DPO?

Section 3 mentions MDP without defining what it refers to.

Eq (3) (4) and (6) seem somewhat redundant. E.g. to me Eq. (6) seems to be Eq. (4) expanded to its token-level and Eq. (6) itself is probably enough for describing the approach and explaining the intuition.

**Questions:**

the authors generate positive & negative examples by sampling. For some of the more challenging reasoning problems it is probably impossible to get a single positive example out of the 100 samples. Can the authors provide the statistics on that?

---

> ### Author Response · Authors · 2024-11-21
> **Response to Official Review by Reviewer 5qbR (1/2)**
>
> Thank you for your valuable feedback. It is true that there are a few flaws in the paper. We have revised the paper based on your suggestions, as detailed below:
>
> **Q1:** The contribution overall seems relatively minor: SC-DPO is not really a new fine-tuning objective but instead the original DPO objective combined with a rather specific way of generating negative examples; the empirical gains are also relatively small.
>
> **A1:** Thank you for your comment. You are correct that SCDPO is not a new learning objective. However, we emphasize our contribution from three aspects.
>
> Firstly, SCDPO represents a novel application of the DPO algorithm, incorporating stepwise information into the DPO process. This is achieved by constructing positive and negative pairs that differ after a certain step. SCDPO leverages the stepwise structure of mathematical reasoning rationales to enhance LLMs' mathematical reasoning abilities through process supervision [1]. Secondly, our method can build the negative and positive pairs by the model itself using temperature adjustment. It is a simple yet effective method for DPO data construction in a low-cost manner. Thirdly, in terms of **empirical gains**, we have summarized the results in the tables below to better demonstrate the effectiveness of SCDPO. SCDPO consistently improves performance across datasets of various types and difficulty levels. Specifically, while InternLM2-SFT-SCDPO only achieves a 0.6% accuracy gain compared to naive DPO on MATH, the out-of-domain datasets OCW and Hungarian show a 4.9-point and 5-point performance gain, respectively, which is significant. It also enhances the performance of three 7B SFT models with varying solution formats.
>
> To further highlight improvements on the 7B models, we have added results from five out-of-domain datasets, which are included in Table 2 of the revised paper.
>
> |Model|Size|GSM8K|MATH|OCW|Hungarian|Mathematics|SVAMP|Simuleq|APE210K|CMATH|MGSM-zh|
> |-|-|-|-|-|-|-|-|-|-|-|-|
> |InternLM2-SFT|20B|86.4|55.8|21.6|71|84.0|86.9|91.2|77.1|88.4|74.8|
> |InternLM2-SFT-DPO|20B|87.0|57.6|25.5|74|85.6|89.7|92.6|78.7|89.9|76.0|
> |InternLM2-SFT-DPO(d-e)|20B|88.2|57.5|24.5|73|86.3|88.9|91.1|78.8|89.3|76.0|
> |InternLM2-SFT-SCDPO|20B|**88.5**|**58.1**|**29.4**|**78**|**87.5**|**90.2**|**93.6**|**79.3**|**90.3**|**80.4**|
>
>
> | **Method**            | **GSM8K** | **MATH** | **OCW** | **Hungarian** | **Mathematics** | **SVAMP** | **Simuleq** |
> |------------------------|-----------|----------|---------|---------------|-----------------|-----------|-------------|
> | **Mistral-7B-Ours**    |           |          |         |               |                 |           |             |
> | SFT (Baseline)         | 76.8      | 43.2     | 21.7    | 52            | 69.8            | 81.3      | 73.9        |
> | SFT-continued          | 76.3      | 43.9     | 18.8    | 55            | 70.3            | 80.8      | 74.5        |
> | SFT+DPO                | 78.8      | 45.1     | 18.4    | 56            | 74.8            | 81.0      | 74.9        |
> | SFT+DPO (d-e)          | 79.0      | 45.7     | 18.0    | 59            | 74.4            | 79.2      | 73.2        |
> | **SFT+DPO+SC**         | **80.1**  | **47.7** | **22.4**| **61**        | **76.5**        | **82.3**  | **79.0**    |
> | **MetaMath-Mistral-7B**|           |          |         |               |                 |           |             |
> | SFT (Baseline)         | 77.7      | 28.2     | 12.5    | 33            | 33.9            | 80.0      | 68.5        |
> | SFT-continued          | 76.8      | 28.5     | 13.2    | 35            | 33.6            | 80.3      | 69.1        |
> | SFT+DPO                | 81.0      | 28.7     | 14.0    | 34            | 33.8            | 81.0      | 68.3        |
> | SFT+DPO (d-e)          | 81.4      | 29.0     | 14.7    | 38            | 34.3            | 80.9      | 70.6        |
> | **SFT+DPO+SC**         | **81.7**  | **29.3** | **15.4**| **42**        | **35.0**        | **81.6**  | **73.2**    |
> | **MathCoder-Mistral-7B**|          |          |         |               |                 |           |             |
> | SFT (Baseline)         | 78.1      | 39.3     | 12.9    | 62            | 70.4            | 79.4      | 80.5        |
> | SFT-continued          | 78.2      | 40.3     | 12.5    | 65            | 71.2            | 77.3      | 80.7        |
> | SFT+DPO                | 79.2      | 42.9     | 14.3    | 65            | 74.9            | 85.4      | 81.3        |
> | SFT+DPO (d-e)          | 78.3      | 41.1     | 14.7    | 68            | 74.9            | 84.9      | 82.3        |
> | **SFT+DPO+SC**         | **80.4**  | **43.4** | **15.7**| **70**        | **75.4**        | **85.4**  | **83.1**    |
>
> [1] Lightman, Hunter, et al. "Let's verify step by step." arXiv preprint arXiv:2305.20050 (2023).

---

> ### Author Response · Authors · 2024-11-21
> **Response to Official Review by Reviewer 5qbR (2/2)**
>
> **Q2:** Most of the information from Table 1. is probably not super important, e.g. why do the authors need to compare the performance of InternLM2-SFT-SCDPO against the listed open-source LLMs? How is this comparison related to SCDPO?
>
> **A2:** Thank you so much for your thoughtful comment.
>
> Firstly, comparing the performance of InternLM2-SFT-SCDPO with the listed open-source LLMs highlights that our method achieves performance comparable to top open-source models, showcasing the effectiveness of SCDPO. To improve the clarity and readability of Table 1, we have simplified it by removing a few less significant baselines in the revised paper. We appreciate your feedback, which helps us refine our presentation.
>
> Secondly, the following table (Table 2 in the paper) further demonstrates that SCDPO consistently outperforms the baseline and naive DPO across different language models, including Mistral, MetaMath, and MathCoder.
>
> | **Method**  | **GSM8K** | **MATH** | **OCW** | **Hungarian** | **Mathematics** | **SVAMP** | **Simuleq** |
> |------|-----------|----------|---------|---------------|-----------------|-----------|-------------|
> | **Mistral-7B-Ours**    |           |          |         |               |                 |           |             |
> | SFT (Baseline)         | 76.8      | 43.2     | 21.7    | 52            | 69.8            | 81.3      | 73.9        |
> | SFT-continued          | 76.3      | 43.9     | 18.8    | 55            | 70.3            | 80.8      | 74.5        |
> | SFT+DPO                | 78.8      | 45.1     | 18.4    | 56            | 74.8            | 81.0      | 74.9        |
> | SFT+DPO (d-e)          | 79.0      | 45.7     | 18.0    | 59            | 74.4            | 79.2      | 73.2        |
> | **SFT+DPO+SC**         | **80.1**  | **47.7** | **22.4**| **61**        | **76.5**        | **82.3**  | **79.0**    |
> | **MetaMath-Mistral-7B**|           |          |         |               |                 |           |             |
> | SFT (Baseline)         | 77.7      | 28.2     | 12.5    | 33            | 33.9            | 80.0      | 68.5        |
> | SFT-continued          | 76.8      | 28.5     | 13.2    | 35            | 33.6            | 80.3      | 69.1        |
> | SFT+DPO                | 81.0      | 28.7     | 14.0    | 34            | 33.8            | 81.0      | 68.3        |
> | SFT+DPO (d-e)          | 81.4      | 29.0     | 14.7    | 38            | 34.3            | 80.9      | 70.6        |
> | **SFT+DPO+SC**         | **81.7**  | **29.3** | **15.4**| **42**        | **35.0**        | **81.6**  | **73.2**    |
> | **MathCoder-Mistral-7B**|          |          |         |               |                 |           |             |
> | SFT (Baseline)         | 78.1      | 39.3     | 12.9    | 62            | 70.4            | 79.4      | 80.5        |
> | SFT-continued          | 78.2      | 40.3     | 12.5    | 65            | 71.2            | 77.3      | 80.7        |
> | SFT+DPO                | 79.2      | 42.9     | 14.3    | 65            | 74.9            | 85.4      | 81.3        |
> | SFT+DPO (d-e)          | 78.3      | 41.1     | 14.7    | 68            | 74.9            | 84.9      | 82.3        |
> | **SFT+DPO+SC**         | **80.4**  | **43.4** | **15.7**| **70**        | **75.4**        | **85.4**  | **83.1**    |
>
>
> **Q3:** Section 3 mentions MDP without defining what it refers to.
>
> **A3:** Thank you very much for bringing this to our attention. MDP stands for Markov Decision Process [1], and we sincerely apologize for any confusion caused. We have clarified this in the revised paper and included the appropriate reference.
>
> [1] Martin L. Puterman. Markov decision processes: Discrete stochastic dynamic programming.
>
> **Q4:** Eq (3) (4) and (6) seem somewhat redundant. E.g. to me Eq. (6) seems to be Eq. (4) expanded to its token-level and Eq. (6) itself is probably enough for describing the approach and explaining the intuition.
>
> **A4:** Thank you so much for your valuable suggestion. In response, we have simplified the equations in the revised paper to enhance clarity. Specifically, we have combined Eq. (4) and Eq. (6) from the original paper to both describe the approach and provide an intuitive explanation. We have retained Eq. (3), which outlines the loss function of DPO, as a preliminary step to ensure the paper remains understandable for readers who may not be familiar with DPO. Thank you once more for your feedback.
>
> **Q5:** the authors generate positive & negative examples by sampling. For some of the more challenging reasoning problems it is probably impossible to get a single positive example out of the 100 samples. Can the authors provide the statistics on that?
>
> **A5:** Thank you for your thoughtful question. When sampling with InternLM-20B-SFT 100 times, we found that 99.8% of the questions in the GSM8K training set and 91.8% of the questions in the MATH training set yielded at least one positive sample. We have added this to Section 2.1 of the revised paper.

---

> ### Author Response · Authors · 2024-11-23
> **Looking forward to your reply**
>
> Dear Reviewer 5qbR,
>
> We hope this message finds you well. As the author-reviewer discussion period progresses, we kindly request your confirmation regarding whether our rebuttal adequately addresses the concerns raised in your review.
>
> We greatly appreciate the time and effort you have dedicated to reviewing our work. Your comments and suggestions have been invaluable in helping us improve our project.
>
> Thank you once again for your thoughtful feedback. We look forward to your further guidance.
>
> Sincerely,
> The Authors

---

> > ### Comment · Reviewer_5qbR · 2024-11-25
> >
> > Thank you for the clarification. I've updated my score accordingly.

---

> > > ### Author Response · Authors · 2024-11-25
> > > **Thank you for your response.**
> > >
> > > Dear Reviewer 5qbR,
> > >
> > > We deeply appreciate you taking the time to review our work and recognize our response and efforts. Your thoughtful feedback and guidance are invaluable to us.
> > >
> > > Thank you once again for the time and effort you have devoted to evaluating our work.
> > >
> > > Best regards,
> > > The Authors

---

### Author Response · Authors · 2024-11-22
**Paper Update Summary**

Thanks to all the reviewers for their constructive suggestions and comments. Below, we summarize the details of the revised paper.

1. Evaluation results of 5 out-of-domain datasets on 7B models are added to Table 2. (Reviewer buSo, Reviewer 1hBn, Reviewer 5qbR)
2. Evaluation results of several important baseline models are added to Table 1. (Reviewer buSo)
3. Definition and reference of MDP (Markov Decision Process) is added to Section 3 on page 4. (Reviewer 5qbR)
4. The unlinked reference in line 466 is fixed in the related work. (Reviewer b65V, Reviewer qGoa, Reviewer 1hBn)
5. Eq. (4) and Eq. (6) are combined into one equation in Section 2.2 on page 4. (Reviewer 5qbR)
6. Statistics of the coverage of correct solutions is added to Section 2.1 on page 3. (Reviewer 5qbR)
7. Comparison with Step-DPO is added to Table 4 and Section 4.2 on page 9, as well as the related work section. (Reviewer b65V, Reviewer 1hBn)
8. Hyperparameters used for generating training data, and the number of samples in the final training data are added to Section 4.1 on page 6. (Reviewer b65V)
9. Typos are fixed: "We" is changed to "we" on page 3. The order of i3 and i2 is switched in Figure 1. (Reviewer buSo)
10. Quantitative and qualitative analysis of the location of actual errors and its relation to the starting point of error generation is added to Appendix E. (Reviewer buSo, Reviewer qGoa)
11. Explanation of the concept of a “step” is added to Section 2 on page 3. (Reviewer qGoa)
12. Results of manipulating the distribution of the step index where error generation begins is added  to Table 7. (Reviewer buSo)

The valuable suggestions and insightful comments from the reviewers have been invaluable in improving this project. We sincerely appreciate the time and effort the reviewers and chairs have dedicated to evaluating our work.

---

### Meta-Review · Area_Chair_nXAe · 2024-12-19

**Metareview:**

This paper presents Step-Controlled DPO (SCDPO), a method that introduces stepwise error supervision into DPO training for improving mathematical reasoning in LLMs. The main idea is to automatically generate negative samples by making the model produce errors at specific steps through temperature adjustment, then using these paired samples for DPO training. The paper demonstrates effectiveness by achieving competitive scores on GSM8K (88.5%) and MATH (58.1%) using InternLM2-20B. The key strengths include: 1) A simple yet effective approach to incorporate stepwise supervision without human annotation; 2) Extensive experiments showing consistent improvements across different models and datasets; 3) Clear presentation and analysis. However, there are several notable weaknesses: 1) The contribution appears relatively minor as it mainly introduces a specific way of data generation rather than a novel training objective; 2) The empirical gains on challenging datasets like MATH are relatively small (0.6% vs naive DPO); 3) Improving LLMs through self-generated data rather than leveraging larger models for data generation presents a fundamental limitation, as the model can only learn from its own generated samples. Given these limitations in technical novelty and empirical results, I recommend rejecting this paper despite its good presentation and systematic evaluation.

**Additional Comments On Reviewer Discussion:**

During the rebuttal period, the reviewers raised several key concerns including: comparison with Step-DPO baseline, effectiveness on out-of-domain datasets, definition of steps, and analysis of error distributions. The authors provided extensive experimental results comparing with Step-DPO showing competitive performance. They also added evaluation on 5 out-of-domain datasets demonstrating good generalization. Additionally, they clarified the step definition and conducted manual analysis of error distributions. While Reviewer 5qbR and Reviewer b65V raised their scores from 3 to 5 after rebuttal, they maintained concerns about the technical novelty and fundamental limitations of self-improvement. Reviewer buSo and Reviewer qGoa kept their positive scores of 8 and 6 respectively, and Reviewer 1hBn raised the score to 6, citing the empirical validation. Weighing these points, while the authors addressed many technical concerns, the core limitations in novelty and methodology remain significant factors in the final rejection decision.

---

### Decision · Program_Chairs · 2025-01-22

Reject